# The lysosomal transporter MFSD1 is essential for liver homeostasis and critically depends on its accessory subunit GLMP

David Massa López[1], Melanie Thelen[2], Felix Stahl[3], Christian Thiel[4], Arne Linhorst[1†], Marc Sylvester[2], Irm Hermanns-Borgmeyer[5], Renate Lüllmann-Rauch[6], Winnie Eskild[7], Paul Saftig[1], Markus Damme[1*]

[1]Institute of Biochemistry, Christian-Albrechts-University Kiel, Kiel, Germany; [2]Institute for Biochemistry and Molecular Biology, Rheinische-Friedrich-Wilhelms-University, Bonn, Germany; [3]Institute of Clinical Chemistry and Laboratory Medicine, University Medical Center Hamburg-Eppendorf, Hamburg, Germany; [4]Center for Child and Adolescent Medicine, Department of Pediatrics I, University of Heidelberg, Heidelberg, Germany; [5]Center for Molecular Neurobiology, University Medical Center Hamburg-Eppendorf, Hamburg, Germany; [6]Institute for Anatomy, Christian-Albrechts-University Kiel, Kiel, Germany; [7]Department of Bioscience, University of Oslo, Oslo, Norway

*For correspondence:
mdamme@biochem.uni-kiel.de

Present address: †Biochemistry I, Department of Chemistry, Bielefeld University, Bielefeld, Germany

Competing interests: The authors declare that no competing interests exist.

**Abstract** Lysosomes are major sites for intracellular, acidic hydrolase-mediated proteolysis and cellular degradation. The export of low-molecular-weight catabolic end-products is facilitated by polytopic transmembrane proteins mediating secondary active or passive transport. A number of these lysosomal transporters, however, remain enigmatic. We present a detailed analysis of MFSD1, a hitherto uncharacterized lysosomal family member of the major facilitator superfamily. MFSD1 is not N-glycosylated. It contains a dileucine-based sorting motif needed for its transport to lysosomes. *Mfsd1* knockout mice develop splenomegaly and severe liver disease. Proteomics of isolated lysosomes from *Mfsd1* knockout mice revealed GLMP as a critical accessory subunit for MFSD1. MFSD1 and GLMP physically interact. GLMP is essential for the maintenance of normal levels of MFSD1 in lysosomes and vice versa. *Glmp* knockout mice mimic the phenotype of *Mfsd1* knockout mice. Our data reveal a tightly linked MFSD1/GLMP lysosomal membrane protein transporter complex.
DOI: https://doi.org/10.7554/eLife.50025.001

## Introduction

Lysosomes are dynamic and membrane-bound organelles ubiquitously found in eukaryotic cells. Their major function is the hydrolytic degradation of macromolecules like proteins, complex lipids, nucleic acids and oligosaccharides after uptake from the extracellular space by pinocytosis and endocytosis or from intracellular sources after fusion with autophagosomes (*Saftig and Klumperman, 2009*). Hydrolytic degradation is mediated by the concerted action of ~60 different, mostly soluble acid hydrolases. The catabolic end-products of these reactions (e.g. amino acids, peptides, monosaccharides, nucleosides) are exported by polytopic transmembrane proteins to the cytosol for anabolic processes. While in the past, the vast majority of soluble proteins of lysosomes were identified due to their deficiencies in lysosomal storage disorders and by proteomics of purified lysosomes

**eLife digest** Lysosomes are specialized, enclosed compartments within cells with harsh chemical conditions where enzymes break down large molecules into smaller component parts. The products of these reactions are then transported out of the lysosome by transporter proteins so that they can be used to build new molecules that the cell needs.

Despite their importance, only a few lysosomal transporters have been thoroughly studied. A protein called MFSD1 had previously been identified as a potential lysosomal transporter, but its precise role has not been described.

Now, Massa López et al. have characterized the role of MFSD1, by genetically modifying mice so they could no longer make the transporter. These mice developed severe liver damage. In particular, a specific type of cell that is important for lining blood vessels in the liver, seemed to be lost in these mice. Older MFSD1 deficient mice also had more tumors in their livers compared to normal mice.

Massa López et al. next examined what happened to other lysosomal proteins in the MFSD1 deficient mice, and found that these mice had strikingly low levels of a protein called GLMP. To better understand the relationship between GLMP and MFSD1, another strain of genetically modified mice was analyzed, this time missing GLMP. Mice without GLMP were found to have very similar liver problems to those observed in the mice lacking MFSD1. Moreover, the GLMP deficient mice had low levels of the MFSD1 protein.

Further experiments demonstrated that MFSD1 and GLMP physically interact with each other: GLMP seemed to protect MFSD1 from being degraded in the harsh internal environment of the lysosome. Thus both GLMP and MFSD1 were needed to form a stable lysosomal transporter.

Characterizing MFSD1 is important for scientists attempting to understand how the lysosomal membrane and transporters work. Moreover, these findings may shed light on how defects in lysosomal transporters contribute to metabolic disease.

DOI: https://doi.org/10.7554/eLife.50025.002

or affinity chromatography and most of them have been characterized extensively, much less is known about the low abundant integral membrane proteins of lysosomes.

The luminal side of the lysosomal membrane is lined with a dense layer of carbohydrates attached to integral membrane proteins, presumed to protect these membrane proteins from the activity of the abundant luminal proteases (*Wilke et al., 2012*). While the highly abundant transmembrane proteins like LAMP1 or LAMP2 are known since decades, there is still a major gap of knowledge about polytopic transporter proteins, mediating the transport of metabolites between the lysosomal lumen and the cytosol, but also the import of metabolites from the cytosol to lysosomes (*Abu-Remaileh et al., 2017*; *Chapel et al., 2013*). Of note, many of such transporters have been biochemically characterized in the 80s and 90s. However, the genes coding for the great majority of those transporters have not been cloned (*Gahl, 1989*; *Pisoni and Thoene, 1991*).

Several proteomics-based studies pointing to identify novel lysosomal membrane proteins have been published (*Bagshaw et al., 2005*; *Chapel et al., 2013*; *Della Valle et al., 2011*; *Schröder et al., 2007*). Chapel et. al. particularly aimed to identify lysosomal transporters and one of their top candidates was 'Major facilitator superfamily domain containing 1' (MFSD1).

MFSD1 is a poorly characterized protein. It belongs to the major facilitator superfamily (MFS) of transporters, one of the two largest family of transporter proteins mediating secondary active or passive transport processes over cellular membranes (*Pao et al., 1998*). MFS transporters move a variety of small compounds across biological membranes. In humans, MFS proteins mediate intestinal nutrient absorption, renal and hepatic clearance, but they have also evolved additional functions in the transport of metabolites and signaling molecules (*Quistgaard et al., 2016*). *MFSD1* is co-expressed in the transcription factor EB (TFEB)-mediated gene network regulating lysosomal biogenesis and lysosomal gene expression and was thus identified as a direct TFEB-target gene (*Palmieri et al., 2011*). Overexpression of epitope-tagged MFSD1 indicated co-localization with LAMP-proteins, demonstrating that it is indeed a resident lysosomal protein (*Chapel et al., 2013*; *Palmieri et al., 2011*). However, there are also reports showing non-lysosomal localization of MFSD1

at the plasma membrane of neurons and the Golgi-apparatus (*Perland et al., 2017*; *Valoskova et al., 2019*).

In this study, we provide a detailed biochemical characterization of MFSD1. Endogenous MFSD1 is localized in lysosomes. It contains 12 transmembrane domains and it is ubiquitously expressed in murine tissues. It harbors a dileucine-based sorting motif in its cytosolic N-terminus which is required for its transport to lysosomes. In order to decipher the physiological function of MFSD1, we generated and analyzed *Mfsd1* knockout (KO) mice. MFSD1-deficient mice develop a severe liver disease characterized by extravasation of erythrocytes, sinusoidal damage, loss of liver sinusoidal endothelial cells (LSECs) and finally signs of fibrosis. By means of differential proteomics of isolated liver lysosomes from wildtype and *Mfsd1* KO mice, we identified GLMP as an essential accessory protein for MFSD1. GLMP is a highly glycosylated lysosomal protein of so far unknown function. Deficiency of *Mfsd1* leads to drastically reduced levels of GLMP and vice versa. MFSD1 and GLMP physically interact and *Glmp*-deficient mice resemble a phenocopy of *Mfsd1* KO mice suggesting the MFSD1/GLMP complex to be a stable and functional relevant lysosomal transporter complex.

## Results

### MFSD1 is a ubiquitously expressed, non-glycosylated polytopic lysosomal membrane protein containing a dileucine-based sorting motif

We and others have identified MFSD1 previously in proteomic analyses of isolated liver lysosomes (*Chapel et al., 2013*; *Markmann et al., 2017*). For validation of its lysosomal localization and the newly generated MFSD1-specific antibodies, we ectopically expressed N- and C-terminally hemagglutinin (HA)-tagged MFSD1 in HeLa cells (*Figure 1A,B*). Co-immunofluorescence staining with antibodies against HA, LAMP2 and MFSD1 confirmed the co-localization of MFSD1 (either detected with HA- or MFSD1 antibodies) with LAMP2 and the specificity of our MFSD1 antibody. In addition to lysosomal localization, staining of the Golgi-apparatus was observed frequently (*Figure 1A*). By immunoblot, both HA- and MFSD1-antibodies detected a major band of ~35 kDa for N- or C-terminally tagged MFSD1 in transfected cells, differing from the predicted molecular weight of ~51 kDa (*Figure 1B*). Untagged MFSD1 was exclusively detected with the MFSD1 antibody (*Figure 1B*, right panel). Additionally, minor bands of smaller molecular weight were detected for all three constructs, suggesting partial proteolysis. Co-immunofluorescence staining of mouse embryonic fibroblasts (MEF) for endogenous MFSD1 with LAMP1 validated the lysosomal localization at the endogenous level (*Figure 1C*) and notably MFSD1 was absent from Golgi-apparatus structures. These data were corroborated by analyzing magnetite-bead isolated lysosomes compared to the postnuclear supernatant from MEFs, showing a pronounced enrichment of endogenous MFSD1 in the lysosome-enriched fraction similar to the striking enrichment of the lysosome-marker LAMP1 (*Figure 1D*). Other organelles were either depleted in these fractions (ER, detected with an antibody against KDEL), or only slightly enriched (mitochondria, detected with an antibody against VDAC and Golgi, detected with an antibody against GM130). We next investigated the tissue-specific expression of MFSD1 using our antibody (*Figure 1E*). MFSD1 was detected ubiquitously in murine organs and highest levels were observed in kidney and spleen. Bioinformatics analysis of the primary sequence of MFSD1 by different online analysis tools predicted variable numbers of transmembrane domains (TMDs). Some tools favored 11 TMD, others predicted 12 TMDs. We, therefore, analyzed if MFSD1 has an even or uneven number of TMDs by selective permeabilization of the PM with digitonin or of the PM and lysosomal membrane with saponin and antibody accessibility of N- or C-terminally HA-tagged MFSD1 (*Figure 1—figure supplement 1A* (I)-(III)). These experiments clearly revealed that MFSD1 presents with an even number of TMDs and likely a 12 TMD spanning topology (*Figure 1F*).

We identified a dileucine-based lysosomal sorting motif close to the cytosolic N-terminus of murine MFSD1 (amino acid 7–12, EDRALL) that is highly conserved among different species and bears similarity to motifs found in MFSD8 and SLC17A5/Sialin, structurally related lysosomal transporters (*Figure 1F,G,H*). Mutation of the two critical leucine residues to alanine (MFSD1[LL/AA]) and transfection of the mutant cDNA in HeLa cells followed by immunofluorescence staining showed a striking missorting of MFSD1 to the plasma membrane (PM) (*Figure 1I*). For a quantitative readout by fluorescence-activated cell scanning (FACS), we inserted an HA-tag in the first luminal loop between TMD1 and TMD2 (between amino acids 76 and 77) to detect MFSD1 at the PM (*Figure 1J*

**Figure 1.** MFSD1 is an ubiquitously expressed, non-glycosylated lysosomal protein that contains a dileucine-based lysosomal sorting motif. (**A**) Co-Immunofluorescence staining of HeLa cells overexpressing N- or C-terminally HA-tagged MFSD1 with antibodies against LAMP2 (red), HA (green) and an antibody against MFSD1 (magenta). (**B**) Immunoblot of HeLa cells transfected with untagged and N- or C-terminally tagged MFSD1 with antibodies against HA (left panel) and MFSD1 (right panel). Gapdh is depicted as loading control. MFSD1-specific bands are labeled with arrow-heads. (**C**) Co-immunofluorescence staining for LAMP1 (red) and endogenous MFSD1 (green) of wildtype MEF cells. (**D**) Immunoblot of the postnuclear supernatant and the lysosome-enriched fraction of magnetite-isolated lysosomes from wildtype MEFs probed with antibodies against MFSD1, LAMP1 (marker for lysosomes), GM130 (marker for the Golgi-apparatus), KDEL (marker for the endoplasmic reticulum) and VDAC (marker for mitochondria). (**E**) Immunoblot of crude membranes of the indicated wildtype mouse tissues probed with the antibody against MFSD1. $N^+/K^+$-ATPase is depicted as a

*Figure 1 continued on next page*

*Figure 1 continued*

loading control. An unspecific band is labeled with an asterisk. (F) Topology model of murine MFSD1. The putative dileucine-based lysosomal sorting motif is indicated. Amino acids confining the TMDs (numbered from 1 to 12) are indicated. (G) Alignment of the dileucine-based lysosomal sorting motifs-containing N-termini of the indicated proteins/species. The critical amino acids constituting the motif are depicted in gray. (H) Web-logo representation of the sequence containing the dileucine motif of MFSD1 orthologues from different species (*Mus musculus, Rattus norvegicus, Homo sapiens, Bos taurus, Gallus gallus, Danio Rerio*). (I) Co-immunofluorescence for LAMP2 (green) and HA (red) of HeLa cells transfected with N-terminally HA-tagged MFSD1$^{LL/AA}$-mutant. (J) (I) Schematic representation of the MFSD1-construct with an internal-HA-tag between TMD 1 and 2 used for FACS. (II) FACS plots of untransfected HeLa cells, cells transfected with wildtype and MFSD1$^{LL/AA}$-mutant with an internal HA tag after staining of HA without permeabilization. (III) Quantification of HA-positive cells from three FACS experiments. ***=p < 0.001. (K) Immunoblot of lysates from MFSD1-HA-transfected HeLa cells treated with PNGaseF or EndoH probed with an antibody against HA. Gapdh and LAMP2 are shown as loading controls and controls for efficient glycosidase-treatment, respectively. (L) Immunoblot of HeLa cells transfected with wildtype C-terminally HA-tagged MFSD1 or the indicated mutants N76Q/N449Q probed with an antibody against HA. Gapdh is shown as loading control.

DOI: https://doi.org/10.7554/eLife.50025.003

The following figure supplement is available for figure 1:

**Figure supplement 1.** Topology of MFSD1 and control/additional experiments validating plasma membrane localization of MFSD1$^{LL/AA}$-mutant.
DOI: https://doi.org/10.7554/eLife.50025.004

(I)). The internal tag did not interfere with intracellular sorting (only minor amounts were mislocalized in the ER) and both constructs were expressed at equal levels (*Figure 1—figure supplement 1C* (I)-(III)). While only subtle amounts of wildtype-MFSD1 with the internal HA-tag were detectable at the PM after transfection of Hela cells and staining for HA (~2% of the cells), a considerably higher number of MFSD1-positive cells was observed after transfection with MFSD1$^{LL/AA}$ (~20% of the cells) (*Figure 1J* (II-III)). Biotinylation of cell-surface proteins followed by streptavidin pull-down of HA-tagged wildtype or MFSD1$^{LL/AA}$ transfected HeLa cells and detection with an HA-antibody showed a similar trend toward higher surface levels of MFSD1$^{LL/AA}$ compared to the wildtype (*Figure 1—figure supplement 1B*). An additional predicted tyrosine-based motif (YGKI, aa 195–198) is, according to our experimentally determined topology, on the luminal side and therefore not accessible for adaptor-protein mediating sorting to lysosomes. Mutating this residue did not show any differences in the localization of MFSD1 as determined by immunofluorescence (not shown) or surface biotinylation (*Figure 1—figure supplement 1C*). These data show that the dileucine motif in the cytosolic N-terminus is critical for the sorting of MFSD1 to lysosomes.

We wondered if MFSD1, similar to the great majority of lysosomal membrane proteins (*Saftig and Klumperman, 2009*), is N-glycosylated. MFSD1 contains two putative N-glycosylation sites (N76 and N449). Protein extracts from HeLa cells transfected with HA-tagged MFSD1 were treated with the endoglycosidases Peptide-N-Glycosidase F (PNGaseF) and Endoglycosidase H (*Figure 1K*). However, none of the two endoglycosidases altered the apparent molecular weight of MFSD1, indicating a lack of N-glycosylation. Additionally, we mutated the critical asparagine residue in the predicted putative N-glycosylation motifs (N76 and N449, respectively) to alanine and analyzed transfected HeLa cell lysates by immunoblot with an antibody against HA (*Figure 1L*). Again, no shift in the apparent molecular weight was observed, demonstrating that MFSD1 is, unusual for a lysosomal membrane protein, not N-glycosylated. In summary, MFSD1 is endogenously localized in lysosomes under steady state conditions, is not N-glycosylated, is ubiquitously expressed in murine tissues and its transport to lysosomes is mediated by a dileucine-based sorting motif in the N-terminus.

### *Mfsd1* knockout mice develop a liver phenotype characterized by liver sinus endothelial cell death and prothrombotic conditions

In order to determine the physiological functions of MFSD1, we generated *Mfsd1* knockout (KO) mice. With targeted embryonic stem (ES) cells obtained from the 'Knockout Mouse Project' consortium, we generated mice expressing a tm1a allele in the germline (*Figure 2A*, *Figure 2—figure supplement 1A*). The expression of *Mfsd1* is abrogated in this strain due to an artificial splice acceptor (SA) site and translation of a LacZ fusion transcript. Expression of the fusion transcript was validated by X-Gal staining of brain sections (*Figure 2—figure supplement 1B*). No residual *Mfsd1*-transcript or -protein was detectable as determined by qPCR, immunoblot, immunohistochemistry in liver lysates or liver sections using our MFSD1 antibody or *Mfsd1*-specific primers (*Figure 2B,C,D*),



**Figure 2.** MFSD1 knockout mice develop focal LSEC degeneration followed by platelet aggregation and splenomegaly. (A) Schematic representation of the tm1a targeting vector containing LacZ and Frt-/Cre recombinase sites. (B) qPCR of total liver mRNA from wildtype and *Mfsd1* KO mice with primers specific for *Mfsd1*. *Mfsd1*-levels are normalized to *Gapdh*. n = 4–5 ***=p < 0.0001. (C) Immunoblot of total liver lysates from wildtype and *Mfsd1* KO mice probed with an antibody against MFSD1. Tubulin is shown as loading control. (D) Immunofluorescence staining of liver sections from wildtype and *Mfsd1* KO mice with an antibody against MFSD1 (white). Nuclei are stained with DAPI (blue). (E) Photomicrographs of the liver and the spleen of wildtype and *Mfsd1* KO mice (age: 14 weeks). (F) Weight of the liver and spleen normalized to total bodyweight of wildtype and *Mfsd1* KO mice (age: 14 weeks). (G) Serum-activity levels of alanine transaminase (ALT), aspartate aminotransferase (AST), glutamate dehydrogenase (GLDH), lactate dehydrogenase (LDH) and albumin from wildtype and *Mfsd1* KO mice (age: 14 weeks). (H) Hematoxylin and Eosin, Toluidine blue and Sirius Red staining of the liver of wildtype and *Mfsd1* KO mice (age: 14 weeks). CV = central vein. (I) Representative electron micrographs of liver sections from wildtype and *Mfsd1* KO mice. (J) Immunofluorescence staining of liver sections of wildtype and MFSD1 KO mice for CD31 (upper panel) and von Willebrand-factor (vWF). (K) qPCR of liver total mRNA from wildtype and MFSD1 KO mice with primers specific for CXCL1, MCP1, CD34, MMP2, MMP9

*Figure 2 continued*

and factor VIII. *=p < 0.05; ***=p < 0.0001. n = 4–5. (L) Immunofluorescence staining of liver sections of wildtype and *Mfsd1* KO mice for Integrin α-IIb. Quantification of the Integrin α-IIb positive area is shown.

DOI: https://doi.org/10.7554/eLife.50025.005

The following figure supplements are available for figure 2:

**Figure supplement 1.** Generation of MFSD1-deficient mice.

DOI: https://doi.org/10.7554/eLife.50025.006

**Figure supplement 2.** Gene ontology classification of differentially expressed proteins using the 'Panther' classification system.

DOI: https://doi.org/10.7554/eLife.50025.007

**Figure supplement 3.** Tie2-conditional mice (*Mfsd1*<sup>flox/flox</sup> Tie2 cre) have the same tuberous liver appearance as the full KO.

DOI: https://doi.org/10.7554/eLife.50025.008

**Figure supplement 4.** Increased frequency of liver tumors at an advanced age (>18 months) in *Mfsd*1 KO mice.

DOI: https://doi.org/10.7554/eLife.50025.009

**Figure supplement 5.** *Mfsd1* knockout mice have normal levels of amino acids in serum and urine and overexpression of PM-localized MFSD1<sup>LL/AA</sup> does not increase the uptake of valine and threonine from acidified cell culture medium.

DOI: https://doi.org/10.7554/eLife.50025.010

implicating that the tm1a allele leads to a complete loss of MFSD1 in this mouse strain (from here on referred as *Mfsd1* KO mice, *Mfsd1*$^{-/-}$). *Mfsd1* homozygous and heterozygous mice were born according to the expected Mendelian frequency (*Figure 2—figure supplement 1C*) and no differences in weight gain were observed in a longitudinal study in male or female mice between wildtype and *Mfsd1* KO mice until 13 weeks-of-age (*Figure 2—figure supplement 1D*). However, both male and female *Mfsd1* KO animals had a reduced body weight with 12 months-of-age compared to wildtype animals (*Figure 2—figure supplement 1E*). While young (<10 weeks-of-age) *Mfsd1* KO animals were unremarkable, the liver of *Mfsd1* KO animals > 12 weeks-of-age revealed a striking pathology characterized by tuberous appearance with uneven surface (*Figure 2E*). The liver weight (normalized to the total bodyweight) was normal, but the spleen showed a significant increase in size of about twofold (*Figure 2F*). Heterozygous animals did not present any abnormalities. Cre-mediated germline deletion of exon 3 in *Mfsd1* yielding the tm1d allele after consecutive breeding of the tm1a mice with constitutively expressing Flp- and Cre-deleter mice (*Figure 2—figure supplement 1F–G*) showed the same macroscopic phenotype, corroborating the finding that tm1a mice are functional knockouts (data not shown). Alanine aminotransferase (ALT), aspartate aminotransferase (AST) and glutamate dehydrogenase (GLDH) values in serum of *Mfsd1* KOs were significantly elevated, but lactate dehydrogenase (LDH) and albumin levels were in the normal range, demonstrating established but generally mild damage of the liver parenchyma (*Figure 2G*). No signs of lysosomal storage or abnormal lysosomes (enlarged lysosomes, aberrant deposition of abnormal luminal storage compounds) were observed at the ultrastructural level in hepatocytes (*Figure 2—figure supplement 1H*). The specific activity of the three lysosomal enzymes β-hexosaminidase, β-galactosidase and β-glucuronidase was not significantly different between wildtype and *Mfsd1* KO mice at 13 weeks (*Figure 2—figure supplement 1I*), further strengthening the finding that lysosomes are not obviously altered. Hematoxylin and eosin and toluidine blue staining of liver sections of *Mfsd1* KO animals of 12 weeks-of-age revealed local foci with hepatocyte atrophy or loss. Ordinary sinusoids in such foci were absent due to destroyed sinusoidal endothelium. The normally wide sinusoids, which in *Mfsd1* wildtype mice were lined by a special fenestrated endothelium (liver sinusoid endothelial cells, LSEC), were in *Mfsd1* KO mice replaced by ordinary narrow non-fenestrated capillaries. The extravasal space was enlarged and the space of Disse was missing. This space is the extracelluar cleft between the microvillous surface of the hepatocytes and the special fenestrated sinusoid endothelium (LSECs), where the exchange processes between hepatocytes and the acellular phase of the blood take place. Hepatocytes in pathologic foci were in close contact both with the extravasal non-circulating blood and the circulating blood in the capillaries. This was indicated by the observation that erythrocytes were found both in extra- and intravasal spaces (*Figure 2H,I*). Sirius Red staining revealed moderate fibrosis particularly pronounced around the focal spots of parenchymal damage (*Figure 2H*). In *Mfsd1* KO mice, LSECs were partially absent in spots of focal damage and hepatocytes had direct contact with their microvilli to the blood. Platelets were frequently observed at sites of damaged sinusoids and EM confirmed the increased deposition of collagen and

pathologic, non-fenestrated capillaries were observed instead of LSECs (*Figure 2I*). Immunofluorescence staining for CD31, a general marker for endothelial cells, revealed a clearly increased staining in the focal spots of liver damage and confirmed the formation of new capillaries (*Figure 2J*). Staining for von Willebrand factor (vWF), an acute-phase protein that initiates thrombocyte-adhesion and binds the sub-endothelial matrix, was strikingly increased, implicating a pro-thrombotic status of the liver. The pro-inflammatory and pro-thrombotic conditions were supported by increased levels of CXCL1, MCP1, CD34, MMP2, and MMP9, all markers of inflammation and endothelial remodeling, as determined by qPCR (*Figure 2K*). Factor VIII, a specific marker for LSECs (*Do et al., 1999*), was significantly downregulated. The pro-thrombotic status and accumulation of platelets was additionally validated by immunofluorescence staining of liver sections for Integrin α-IIb/CD41, a protein highly expressed in megakaryocytes and platelets (*Figure 2L*). Platelets were observed frequently particularly in areas of sinusoidal remodeling. A quantitative TMT-based proteomics experiment from total liver lysates of wildtype and *Mfsd1* KO mice (*Figure 2—figure supplement 2A,B*, *Supplementary file 1*) additionally supported our results of endothelial remodeling, platelet aggregation and pro-fibrotic conditions obtained by qPCR and histology. Gene ontology classification of proteins that showed statistically significant differences (p<0.05) using the 'Panther'-classification system (*Thomas et al., 2003*) revealed a clear enrichment of a number of related pathways like 'lymphocyte-aggregation' and 'leukocyte aggregation', 'platelet-aggregation', 'blood coagulation' and 'wound healing' (*Figure 2—figure supplement 2C*). We wondered if the LSEC degeneration is due to a LSEC-cell-intrinsic trigger or caused for example as a secondary alteration due to dysfunctional hepatocytes, and therefore conditionally deleted *Mfsd1* in endothelial cells by crossing floxed *Mfsd1* mice (*MFSD1^tm1c*) with Tie2-cre deleter mice (*Figure 2—figure supplement 3A–C*). Tie2-conditional mice (*Mfsd1^flox/flox Tie2 cre*) lacked expression of MFSD1 in LSECs but not in hepatocytes (*Figure 2—figure supplement 3D*), and notably had the same tuberous liver appearance as the full germline *Mfsd1* KO, strengthening a pivotal role of MFSD1 in LSECS (*Figure 2—figure supplement 3E,F*). It should be noted, however, that Tie2 cre is also expressed in Kupffer cells, and additionally deleted *Mfsd1* in Kupffer cells. In summary, the knockout of *Mfsd1* leads to a focal damage of sinusoids characterized by loss of LSECs and new capillarization. Likely as a consequence, a pro-thrombotic cascade is initiated leading to the accumulation and aggregation of platelets at sites of LSEC loss and sinusoidal damage. Most *Mfsd1* KO mice reach a normal life span, and the liver phenotype does not show a major deterioration. However, we observed a significantly higher occurrence of liver tumors in old animals (>1.5 years), likely as a consequence of fibrosis (*Figure 2—figure supplement 4A,B*).

In a first effort aiming to identify MFSD1 substrates and assuming that a toxic metabolite might be increased in the circulation, we analyzed the blood and urine from *Mfsd1* KO mice for different amino acids (*Figure 2—figure supplement 5A,B*). MFSD1 shows a relatively high homology to members of the proteobacterial intraphagosomal amino acid transporter (Pht) family, transporters found in *Legionella pneumophila* transporting valine and threonine (*Sauer et al., 2005*). However, no major difference for any of the analyzed amino acids was detectable between wildtype and *Mfsd1* KO mice in either serum or urine. A direct uptake of a radioactively $^3$H-labeled amino acid mixture, $^3$H threonine or $^3$H valine from acidified cell culture medium in HEK-cells overexpressing the MFSD1$^{LL/AA}$ construct was also undetectable (*Figure 2—figure supplement 5C,D*). HEK-cells transfected with LYAAT, the lysosomal transporter for small neutral amino acids, efficiently accumulated $^3$H proline, included in our experimental setup as a positive control, and $^3$H labeled amino acids from the amino acid mixture. These data indicate that MFSD1 is likely not a transporter for any of the tested amino acids at least under the tested conditions.

## MFSD1 deficiency leads to strikingly decreased levels of the lysosomal membrane protein GLMP

We next investigated how the deficiency of MFSD1 affects the composition of the lysosomal proteome. Lysosomes from three wildtype and three *Mfsd1* KO mice were isolated from the liver by differential centrifugation followed by sucrose-gradient centrifugation after injection of the mice with Tyloxapol. This enrichment typically yields fractions ~ 40–50 fold enriched for the lysosomal marker enzyme β-hexosaminidase (*Damme et al., 2010*; *Markmann et al., 2017*). The lysosome-enriched fractions were used for isotopic labeling-based differential quantitative proteomics (*Figure 3A*). A number of proteins showed slight but significant differences between wildtype and *Mfsd1* KO mice (*Figure 3B,C Supplementary file 2*) but only the levels of two proteins showed outstanding

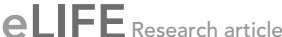

**Figure 3.** Deficiency of MFSD1 leads to specific depletion of GLMP. (A) Scheme of the experimental workflow for subcellular fractionation and purification of lysosomes from the liver followed by Tandem Mass Tag (TMT)- based differential proteomics. (B) Volcano-plot representation of differential protein levels of isolated liver lysosomes from wildtype and *Mfsd1* KO mice determined by mass spectrometry. n = 3. (C) Full list of proteins with differential expression levels between wildtype and *Mfsd1* KO mice with a fold change >1.5 (blue)/−1.5 fold (red) and a p-value<0.05. (D) Immunoblot of isolated liver lysosomes from wildtype and *Mfsd1* KO mice with antibodies against GLMP, MFSD1 and LAMP1. A quantification of the GLMP levels normalized to LAMP1 levels is depicted. (E) Quantification of the immunoblot depicted in (D). The ratio between the MFSD1 signal and the LAMP1 signal is shown. (F) Immunoblot of crude membrane extracts of the indicated tissues from wildtype and *Mfsd1* KO mice for GLMP and MFSD1. $N^+/K^+$-ATPase is depicted as a loading control. An unspecific band is labeled with an asterisk. (G) A quantification of the immunoblot-signals for the GLMP levels in liver normalized to $N^+/K^+$-ATPase levels is depicted ***=p < 0.0001; n = 3. (H) qPCR of liver total mRNA from wildtype and *Mfsd1* KO mice with primers specific for *Glmp*. *Glmp* transcript levels are normalized to *Gapdh* expression and expressed as fold change compared to the wildtype. ***=p < 0.0001; n = 3.

DOI: https://doi.org/10.7554/eLife.50025.011

differences: MFSD1 was strikingly decreased as expected, but additionally another protein showed prominently reduced levels in all three replicates: GLMP, formerly known as NCU-G1 and later renamed to 'Glycosylated Lysosomal Membrane Protein' (*Kong et al., 2014*; *Schieweck et al., 2009*). GLMP is a highly glycosylated type I transmembrane protein previously shown to be a

component of the lysosomal membrane (*Schieweck et al., 2009*). In order to validate our proteomics-based findings, we detected GLMP by immunoblot with an antibody raised against its luminal domain (*Figure 3D,E*). In confirmation with our proteomics data, GLMP was undetectable in isolated liver lysosomes from *Mfsd1* KO mice. We next investigated if GLMP levels are likewise reduced in total liver membranes and moreover also in other tissues of *Mfsd1* KO mice. GLMP levels were strongly diminished in all tested tissues including kidney, liver, lung and spleen (*Figure 3F,G*). Notably, qPCR of total liver cDNA revealed comparable levels of the *Glmp* transcript between wildtype and *Mfsd1* KO animals (*Figure 3H*), indicating a posttranscriptional mechanism leading to GLMP reduction. In summary, *Mfsd1* KO leads to a specific and almost complete absence of GLMP.

## GLMP is essential for maintenance of MFSD1 levels and *Glmp* KO mice resemble the phenotype of *Mfsd1* KO mice

The apparent lack of GLMP in tissues of *Mfsd1* KO mice implicates that both proteins directly interact, depend on each other or regulate each other. We therefore tested, if the absence of GLMP conversely leads to altered MFSD1 levels. Immunoblotting of membrane fractions from *Glmp* KO mice (*Kong et al., 2014*) revealed an almost complete absence of MFSD1 in all analyzed tissues (brain, heart, kidney, liver, lung and spleen) comparable to the reduction of GLMP in *Mfsd1* KO mice (*Figure 4A,D*). Immunofluorescence staining of liver and kidney sections additionally confirmed the absence of any immunoreactive MFSD1 in *Glmp* KO tissue (*Figure 4B*). Similar to the *Glmp* transcript levels in *Mfsd1* KO mice, *Mfsd1* transcript levels were unchanged in *Glmp* KO mice, again pointing towards posttranscriptional regulatory mechanism (*Figure 4C*). Interestingly, the liver of *Glmp* KO mice has, like *Mfsd1* KO mice, a tuberous appearance with uneven surface and fibrosis (*Kong et al., 2014*). We therefore analyzed the liver of *Glmp* KO mice in more detail to directly compare the phenotype with that of *Mfsd1* KO mice. Similar to *Mfsd1* KO mice, *Glmp* KO mice showed local foci where hepatocytes were lost and LSEC were replaced by ordinary capillary endothelium (*Figure 4E*). Similar to *Mfsd1* KO mice, the sites of liver injury in *Glmp* KO mice showed increased staining for CD31 and vWF at sites of liver injury (*Figure 4F*). Finally, we determined the same set of transcripts as we did for *Mfsd1* KO mice. With the exception of Factor VIII, all evaluated transcripts showed the same differential regulation as in *Mfsd1* KO mice (*Figure 4G*). It should be noted that *Glmp* KO mice, similar to *Mfsd1* KO mice, suffer from splenomegaly (*Kong et al., 2014*). In conclusion, GLMP deficiency leads to a drastic decrease of MFSD1 and *vice versa*, and both *Glmp*- and *Mfsd1* KO mice present a phenocopy as a result of 'co-deficiency' of both proteins.

## MFSD1 and GLMP physically interact and GLMP is necessary for the stability of MFSD1 in lysosomes

The genetic data from *Mfsd1* KO and *Glmp* KO mice provide strong evidence for a direct dependence of both proteins on each other. We therefore tested next, if both proteins physically interact using co-immunoprecipitation experiments (*Figure 5A*). MFSD1 could be efficiently immunoprecipitated from HeLa cells transfected with HA-tagged GLMP and untagged MFSD1 with the MFSD1-specific antibody and GLMP-HA was co-immunoprecipitated. Vice versa, GLMP-HA was efficiently precipitated with the HA antibody and MFSD1 was co-precipitated. Isotype-matched control-antibodies neither precipitated GLMP-HA nor MFSD1. LAMP1-HA, although expressed with lower efficiency compared to GLMP-HA, included as a negative control, was not co-immunoprecipitated with MFSD1 and not detected even after longer exposure. It should be noted that the co-immunoprecipitation was highly dependent on the choice of the detergent, and any other tested detergent than CHAPS failed to keep the interaction intact (*Figure 5—figure supplement 1A*). We additionally repeated the co-immunoprecipitation experiments with GFP-tagged MFSD1 and HA-tagged GLMP and included additional controls: C-terminally GFP-tagged MFSD1 was efficiently precipitated using the GFP antibody, and HA-tagged GLMP was efficiently co-immunoprecipitated, but not HA-tagged LAMP1 included as a negative control (*Figure 5—figure supplement 1B*). Endogenous LIMP2, another unrelated lysosomal membrane protein, was efficiently detected in the lysates, but not in the precipitate with the GFP antibody, providing additional evidence for a specific immunoprecipitation. Just like the wildtype proteins, PM localized MFSD1$^{LL/AA}$-GFP and GLMP$^{Y400A}$ where co-immunoprecipitated though with lower efficiency, indicating that the two mutants used for the amino-acid uptake experiments (*Figure 2—figure supplement 5A-D*) still interact and form the complex which



**Figure 4.** GLMP is essential for the maintenance of MFSD1 levels. (**A**) Immunoblot of crude membrane extracts of the indicated tissues from wildtype and *Glmp* KO mice with antibodies against MFSD1 and GLMP. $N^+/K^+$-ATPase is depicted as a loading control. An unspecific band is labeled with an asterisk. Quantification of the MFSD1 levels in liver normalized to $N^+/K^+$-ATPase levels is depicted ***=p < 0.0001; n = 3. (**B**) Immunofluorescence staining of liver- (upper panel) and kidney-sections (lower panel) of wildtype and *Glmp* KO mice for MFSD1 (red) and LAMP1 (green). Nuclei are stained

Figure 4 continued

with DAPI (blue). (**C**) A quantification of the immunoblot-signals for the MFSD1 levels in liver normalized to N$^+$/K$^+$-ATPase levels is depicted ***=p < 0.001; n = 3. (**D**) qPCR of liver total mRNA from wildtype and *Glmp* KO mice with primers specific for *Mfsd1*. *Mfsd1* transcript levels are normalized to *Gapdh* expression and expressed as fold change compared to the wildtype. ***=p < 0.0001; n = 3. (**E**) Hematoxylin and Eosin and Toluidine blue stainings and representative electron microscopy of liver sections of wildtype and *Glmp* KO mice (age: 14 weeks). (**F**) Representative immunofluorescence stainings of CD31 (green) and von Willebrand factor (vWF) (red) of liver sections from wildtype and GLMP KO mice. (**G**) qPCR of liver total mRNA from wildtype and *Glmp* KO mice with primers specific for *Cxcl1, Mcp1, Cd34, Mmp2, Mmp9* and *Factor VIII*. *=p < 0.05; ***=p < 0.0001. n = 4–5.

DOI: https://doi.org/10.7554/eLife.50025.012

might be necessary for transporter activity (*Figure 5—figure supplement 1C*). In summary, these experiments suggest a direct physical interaction of GLMP and MFSD1 and support our genetic data.

In an independent experimental setup, we validated the physical interaction between MFSD1 and GLMP by FACS-based fluorescence resonance energy transfer (FRET) (*Figure 5B–C*). As FRET couples we used the red fluorescent protein mKATE2 and the green fluorescent protein eGFP, which were previously shown to yield efficient FRET (*Fettelschoss et al., 2017*). Both, MFSD1 and GLMP were fused either to eGFP or mKate2 and the constructs were transfected in HeLa cells. Both fusion proteins properly localized to lysosomes as determined by immunofluorescence staining. As negative controls the polytopic lysosomal transporter MFSD8 and the single-transmembrane protein LAMP1 were included in the experimental setup. As positive control for FRET, a fusion protein between eGFP and mKATE2 was transfected. The eGFP-fusion proteins were expressed to similar extent in all samples (*Figure 5—figure supplement 1D*). However, notably only the combination of MFSD1 and GLMP, independent of which tag was used, yielded efficient FRET comparable to the positive control of the eGFP-mKATE2 fusion protein, but not any of the negative controls, indicating close proximity of the two proteins and thereby their physical interaction.

In order to decipher the interplay between MFSD1 and GLMP mechanistically, we established a cell-based model. MEFs from both KO mouse lines were generated. Analysis of the MEF lines by immunocytochemistry and LysoTracker staining did not reveal any differences in regard to the size of lysosomes (determined by LAMP1 staining), delivery of the lysosomal enzyme cathepsin D or their pH, as determined by LysoTracker staining (*Figure 6—figure supplement 1A*). We analyzed GLMP and MFSD1 expression levels in membrane extracts by immunoblot (*Figure 6A*). GLMP levels were below the limit of detection in *Mfsd1* KO MEFs. MFSD1 on the other hand was strongly reduced in *Glmp* KO MEFs, but not completely absent as observed before in most murine *Glmp* KO tissues (*Figure 3A,B*). qPCR of MEFs revealed an increase of the transcripts of *Mfsd1* in *Glmp* KO MEFs and an increase of the *Glmp* transcript in *Mfsd1* KO MEFs (*Figure 6B*), suggesting a transcriptional upregulation as a compensation for the reduced protein levels. We next analyzed the localization of the remaining MFSD1 in *Glmp* KO MEFs (*Figure 6C*). *Mfsd1* KO MEFs were used as a specificity control for the MFSD1-antibody. While MFSD1 shows extensive co-localization with LAMP1 in lysosomes of wildtype MEFs, MFSD1 staining was absent in *Mfsd1* KO MEFs validating the specificity of the staining. The remaining amounts of MFSD1 in *Glmp* KO MEFs notably did not co-localize with LAMP1, but with the Golgi marker GM130. A rescue experiment using transfection of HA-tagged GLMP fully restored the co-localization of MFSD1 with LAMP1 in *Glmp* KO MEFs, while HA-tagged LAMP1, used as a negative control, did not rescue MFSD1, indicating a critical interdependence of GLMP and MFSD1. This experiment excludes the possibility of clonal differences of the cell lines as an explanation for the Golgi-localization of MFSD1 in *Glmp* KO cells (*Figure 6D*). These data also implicate that the interaction of MFSD1 and GLMP is essential for the stability of both proteins, but might additionally affect post-Golgi-sorting.

We therefore analyzed if either MFSD1 or GLMP are properly transported to lysosomes in MEFs with a KO of the other interaction partner. Transfection of HA-tagged MFSD1 in *Glmp* KO MEFs followed by immunofluorescence staining for HA and LAMP1 revealed predominantly co-localization between MFSD1 and LAMP1, indicating proper sorting to lysosomes (*Figure 6E*). *Vice versa*, expression of HA-tagged GLMP in *Mfsd1* KO MEFs similarly showed predominantly lysosomal localization. These data suggest that the interaction between MFSD1 and GLMP is not entirely essential for their

**Figure 5.** MFSD1 and GLMP physically interact. (**A**) Co-immunoprecipitation of untagged MFSD1 and HA-tagged GLMP from transfected HeLa cell lysates with antibodies against HA and MFSD1. Isotype-matched primary antibodies (rb = rabbit) where used for immunoprecipitation as negative-controls. LAMP1 and GLMP were detected with the HA antibody, MFSD1 was detected with the MFSD1 antibody. (**B**) Flow cytometry-based FRET analysis of HeLa cells transfected with combinations of plasmids coding for MFSD1, GLMP, LAMP1 or MFSD8 tagged to either eGFP or mKATE2. FRET

*Figure 5 continued on next page*

*Figure 5 continued*

intensity is plotted against eGFP intensity. Cells inside the gate defined by the intensity of cells expressing fused eGFP:mKATE2 were considered FRET[+]. The number in the plot represents the average of the FRET[+] cells alive for each condition, which is also represented in the bar graph. (**C**) Immunofluorescence of HeLa cells transfected with GLMP fused to mKATE2 (red) and MFSD1 fused to GFP (green). Both fusion proteins show a high degree of co-localization.

DOI: https://doi.org/10.7554/eLife.50025.013

The following figure supplement is available for figure 5:

**Figure supplement 1.** Screening of different detergents for the MFSD1-GLMP-interaction by co-immunoprecipitation and additional controls for FACS-based FRET and immunoprecipitation.

DOI: https://doi.org/10.7554/eLife.50025.014

## GLMP protects MFSD1 from lysosomal proteolysis

Finally we evaluated if a blockage of lysosomal proteolysis can rescue MFSD1 levels in *Glmp* KO MEFs (*Figure 6F*). Treatment of GLMP wildtype and *Glmp* KO MEFs with Bafilomycin A (an inhibitor of lysosomal acidification), E64D or Leupeptin (both inhibitors of lysosomal proteases) partially restored the levels of MFSD1 as determined by immunoblot, indicating that accelerated lysosomal proteolysis causes the decrease in MFSD1 levels in the absence of GLMP. These data were supported by transfection of HA-tagged MFSD1 in HeLa cells alone, or co-transfection with GLMP or LAMP1 followed by immunoblot for HA and MFSD1 (*Figure 6G*). Expression of MFSD1-HA alone and co-transfection with LAMP1 leads to the generation of N- and C-terminal proteolytic fragments (see also *Figure 1B*). Treatment of cells with different inhibitors for lysosomal proteolysis (Chloroquine, $NH_4Cl$, Bafilomycin A and E64D) significantly reduced the formation of these fragments (*Figure 6—figure supplement 1B*), indicating that they are generated by lysosomal proteases. Notably, co-transfection of MFSD1-HA with GLMP, but not LAMP1, significantly reduced the formation of these proteolytic fragments, indicating that specifically GLMP is able to protect MFSD1 from proteolysis (*Figure 6G*). In line with these experiments, immunoblot analysis of cells transfected with the wildtype MFSD1-HA cDNA, but not the PM-localized MFSD1[LL/AA]–HA mutant showed the proteolytic fragments (*Figure 6—figure supplement 1C*), indicating that they are not due to misfolding or destabilization of MFSD1 in pre-lysosomal compartments, but occur specifically in lysosomes if GLMP is limiting. In conclusion, MFSD1 and GLMP physically interact, their transport to lysosomes does not essentially depend on each other and the stability MFSD1 against lysosomal proteolysis depends on GLMP. A schematic representation of the two interacting partners is shown in *Figure 6H*.

## Discussion

The understanding of the lysosomal proteome has considerably improved in the past years. However, there is still a major gap of knowledge about the nature of polytopic transporter proteins, mediating the transport of metabolites between the lysosomal lumen and the cytosol. This is surprising, given the lack of knowledge of the lysosomal transporters for most amino acids, peptides, monosaccharides and several other small solutes. Previous proteomic studies identified a large number of novel putative lysosomal transporters (*Chapel et al., 2013*), and in this study, we characterized one of those candidate transporters, MFSD1 in detail. We identified GLMP as an essential accessory subunit of MFSD1 and establish an essential role of GLMP and MFSD1 in the homeostasis of the liver and particularly LSECs.

The major goal of this study was to identify the substrate(s) of MFSD1 by analyzing KO mice, assuming that a substrate accumulates in lysosomes. The phenotype of *Mfsd1* KO mice is complex and the liver phenotype did not resemble that observed in mouse models of lysosomal storage diseases. We failed to detect any signs of typical lysosomal storage like aberrant storage material or enlarged lysosomes in the liver or MEFs. By comparing the liver and spleen pathology with other pathological situations, especially involving the liver sinusoids, we found some similarities of the phenotype observed in *Mfsd1* KO mice (and *Glmp* KO mice) with 'sinusoidal obstruction syndrome'



**Figure 6.** The interaction between MFSD1 and GLMP is dispensable for trafficking to lysosomes and GLMP protects MFSD1 from proteolytic degradation. (**A**) Immunoblot of crude membrane extracts of wildtype, *Glmp* KO and *Mfsd1* KO MEFs with antibodies against GLMP and MFSD1. N+/K+-ATPase is depicted as a loading control. A quantification of the GLMP and MFSD1 levels in MEFs normalized to N+/K+-ATPase levels is depicted ***=p < 0.0001; n = 3. **=p < 0.001; n = 3. (**B**) qPCR of MEF total mRNA from wildtype, *Glmp* KO and *Mfsd1* KO mice with primers specific for *Mfsd1*

*Figure 6 continued*

and *Glmp*. *Mfsd1/Glmp* transcript levels are normalized to *Gapdh* expression and expressed as fold change compared to the wildtype. \*\*\*=p < 0.0001; n = 3. (C) Co-immunofluorescence of wildtype, *Glmp* KO and *Mfsd1* KO MEFs for endogenous MFSD1 (green) together with LAMP1 (upper panel, red) and the Golgi-marker GM130 (lower panel; red). (D) Co-immunofluorescence staining of MEFs from wildtype and *Glmp* KO mice transfected with HA-tagged GLMP (upper panel) or HA-tagged LAMP1 (lower panel) for endogenous MFSD1 (green) together with LAMP1 (magenta) and antibody against HA (red) to detect HA-tagged GLMP and HA-tagged LAMP1. An untransfected cell is labeled with an asterisk. (E) Co-immunofluorescence staining of MEFs from *Glmp* KO and *Mfsd1* KO mice ectopically overexpressing HA-tagged MFSD1 or GLMP, respectively, for HA (red). Endogenous LAMP1 (green) is stained as a lysosomal marker. (F) Immunoblot analysis for MFSD1 of wildtype and *Glmp* KO MEFs treated with the indicated inhibitors (BafA = Bafilomycin A; E64D, Leu = Leupeptin) for 16 hr. Tubulin is depicted as a loading control. (G) Immunoblots of lysates from HeLa cells transfected with HA-tagged MFSD1 alone, co-transfected with GLMP or co-transfected with LAMP1 with antibodies against the HA-tag, MFSD1, GLMP and LAMP1. Gapdh is depicted as a loading control. MFSD1-specific bands are labeled with arrow-heads. A quantification of the blots of the proteolytic fragments is depicted (\*\*\*=p < 0.0001; \*\*=p < 0.001, \*=p < 0.05; n = 3). (H) Schematic representation of the topology of the two interacting proteins MFSD1 (blue) and GLMP (orange).

DOI: https://doi.org/10.7554/eLife.50025.015

The following figure supplement is available for figure 6:

**Figure supplement 1.** Treatment with lysosomal inhibitors decrease the generation of the MFSD1 C-terminal fragment.

DOI: https://doi.org/10.7554/eLife.50025.016

(SOS), a pathology observed in humans and mouse models treated with oxaliplatin (an adjuvant or neoadjuvant for chemotherapy for colorectal carcinoma) or patients taking pyrrolizidine alkaloid-containing herbal remedies (*DeLeve et al., 2002*; *Fan and Crawford, 2014*). A central pathological event under these conditions is a toxic destruction of hepatic sinusoidal endothelial cells, with sloughing and downstream occlusion of terminal hepatic venules. Extravasated platelet aggregation in the space of Disse (as observed in *Mfsd1* KO and *Glmp* KO mice) was observed in a rat model for SOS (*Hirata et al., 2017*). Contributing factors are LSEC glutathione- and nitric oxide depletion. Since the LSECs are in extensive exchange with the blood, it is worthwhile speculating that a toxic metabolite present in trace amounts, possibly transported under normal conditions by MFSD1, is circulating in the bloodstream and causing LSEC cell death. However, while it is reasonable to assume that the accumulation of a toxic substrate or dysfunction of lysosomes due to the loss of the transporter function of MFSD1 is the underlying cause for the observed severe phenotype in *Mfsd1* and *Glmp* KO mouse strains, we cannot provide final certainty for this hypothesis. We cannot finally exclude that GLMP might also interact with other (lysosomal membrane) proteins or provides general stability for the lysosomal membrane like LAMP proteins do. MFSD1 might be important for indirectly transporter-related functions like nutrient sensing or even transporter-unrelated processes.

MFSD1 shows a relatively high homology to members of the proteobacterial intraphagosomal amino acid transporter (Pht) family, transporters found in *Legionella pneumophila* transporting valine and threonine (*Sauer et al., 2005*). However, we did not observe any major difference in the serum of *Mfsd1* KO mice for any of the analyzed amino acids and failed to show a valine- or threonine transport in a cell-based assay. While these results do not exclude amino acids as substrates, other metabolites are more likely transported by MFSD1. The identification of the substrate(s) will be the key for understanding the observed liver pathology and might reveal important understanding of the SOS-like phenotype.

Overexpressed human and mouse MFSD1 was previously shown to co-localize with LAMP1 or LAMP2, respectively, indicating that it is a resident lysosomal protein (*Chapel et al., 2013*; *Palmieri et al., 2011*). However, other reports revealed a non-lysosomal localization of MFSD1 at the PM of neurons and the in Golgi-apparatus (*Perland et al., 2017*; *Valoskova et al., 2019*). Our experiments, detecting endogenous MFSD1 in mouse cells and tissues with perfect co-localization with lysosomal markers leave no doubt on its localization in lysosomes. We can exclude any antibody-related unspecificity issues due to the inclusion of knockout samples. However, interestingly we also observed Golgi-localized MFSD1 upon overexpression in HeLa cells, but not at the endogenous level in MEF cells. A likely explanation for this discrepancy is that GLMP is a limiting factor under these overexpression conditions.

MFSD1, together with other MFS transporters, belongs to the atypical SLCs. Most MFS transporters are similar in structure, despite their relatively low sequence identities (*Reddy et al., 2012*). The

assignment to the MFS-clan suggested a 12 transmembrane topology, but here we provide experimental proof for this topology.

Extensive glycosylation is a typical feature of the great majority of lysosomal proteins including the integral membrane proteins. The abundant presence of N-glycans results in a tight glycocalyx which is thought to protect lysosomal membrane proteins from degradation by luminal lysosomal proteases. According to our data, MFSD1 is not N-glycosylated. MFSD1 is therefore one of the very few known non-glycosylated multi-transmembrane spanning integral lysosomal membrane proteins together with few rare exceptions: The chloride channel/Cl(-)/H(+)-exchanger CLC-7, similarly lacks N-glycosylation. Of interest CLC-7 was shown to act in a complex with a single-transmembrane spanning highly glycosylated type I transmembrane protein OSTM1. OSTM1 stabilizes CLC-7 but is also essential for mediating the trafficking of the CLC-7/OSTM1 complex from the ER to lysosomes (*Lange et al., 2006*). Finally, OSTM1 critically regulates Cl(-)/H(+)-exchange (*Leisle et al., 2011*). However, the interaction between MFSD1 and GLMP differs at least in one regard: GLMP and MFSD1 can reach lysosomes independent from each other, as shown by expression of both proteins in the corresponding KO MEFs. If the interaction is also critical for the transporter activity of MFSD1 remains to be determined.

In regard to the dependence on an accessory subunit, MFSD1 and CLC-7 closely resemble a number of plasma-membrane localized transporter proteins which contain β-subunits. The prototypical examples are the Na,K-ATPase or voltage gated potassium channels (*Geering, 2008*; *Pongs and Schwarz, 2010*). Even though for ion channels and members of the ABC transporter family β-subunits are very common, accessory subunits are not that common for members of the SLC-transporter family. One of those SLCs is the monocarboxylate transporter-1 (MCT1). The correct translocation of MCT-1, a PM-localized transporter for monocarboxylates such as L-lactate and pyruvate, critically depends on the interaction with basigin or embigin, both highly glycosylated type I transmembrane proteins acting as accessory subunits (*Halestrap, 2012*). Interestingly, both bear two or three extracellular immunoglobulin domains, respectively. A secondary structure prediction of the luminal domain of GLMP using the 'Phyre2'-web portal (*Kelley et al., 2015*) predicts similarity with the immunoglobulin-fold, and it can be speculated that the interaction of SLC transporters and immunoglobuline-like proteins have developed co-evolutionary.

Very recently, another lysosomal SLC-transporter was shown to critically depend on an accessory subunit: the nitrogen-containing-bisphosphonate transporter SLC37A3 forms a complex with the type I single-transmembrane spanning protein ATRAID (all-trans retinoic acid-induced differentiation factor) (*Yu et al., 2018*). Very much like in the situation of MFSD1 and GLMP, the knockout of *ATRAID* leads to highly reduced stability of the transporter SLC37A3, implicating that ATRAID confers protection to SLC37A3. Notably, similar to OSTM1 and GLMP, ATRAID has a highly glycosylated luminal N-terminus and a short cytosolic C-terminus. These examples implicate, that the formation of lysosomal complexes between a transporter and an accessory subunit might be much more common than anticipated. SLC37A3 itself is glycosylated. Although the intracellular sorting of ATRAID in *SLC37A3* KO cells and the sorting of SLC37A3 in *ATRAID* KO cells was not investigated, SLC37A3 is hypoglycosylated upon ectopic expression in *ATRAID* KO cells, implicating at least in part an important role of the ATRAID in intracellular sorting (*Yu et al., 2018*).

Most accessory-/ β-subunits are essential for the stability of their corresponding transporters, but additionally mediate at least in part the intracellular sorting of their corresponding polytopic transmembrane interaction partner. Our data show that the sorting of ectopically expressed GLMP in *Mfsd1* KO cells or MFSD1 in *Glmp* KO cells to lysosomes is not critically altered, but that instead the stability against lysosomal proteolysis of MFSD1 is altered in *Glmp* KO cells. While we cannot exclude subtle alterations in the sorting, the protection of MFSD1 against proteolysis seems to be more critical. On the other hand, while this explanation sounds reasonable for MFSD1, the levels of GLMP in *Mfsd1* KO cells are also strikingly reduced, and the stability of GLMP apparently also critically depends on the non-glycosylated MFSD1, a counter-intuitive finding that implicates that not only glycosylation matters, but also the protein-protein interaction is essential to maintain a stable protease-resistant complex. It would be of great interest to determine the interaction in more detail, and which protein domains/loops of GLMP and MFSD1 convey this interaction. Additionally GLMP might be important for promoting a more stable conformation of MFSD1, and it would be interesting to figure out if the two proteins already interact in the ER. Once a natural substrate for MFSD1 is identified, a major question is of course if the interaction with GLMP is also essential for the

transport, for example by mediating substrate recognition or increase substrate supply. Unlike channels or enzymes that are relatively static, solute carriers (to which the MFS-family belongs to) must undergo rapid conformational changes during their functions. Therefore, the presence of a stable globular protein domain may affect the conformational dynamics of these solute carriers and directly regulate its transporter activity.

## Materials and methods

### Antibodies and reagents

An MFSD1-specific polyclonal antibody was generated by immunizing rabbits with the peptide 'EDEDGEDRALLGGRREADC' corresponding to the N-terminus of mouse MFSD1. Serum was collected 150 days after the first immunization. A GLMP-specific polyclonal antibody was generated by immunizing rabbits with the peptide 'CQAFSRSGRPAQPPRLLH' within the luminal domain of mouse GLMP. Both antisera were affinity-purified against the corresponding immunization-peptides. MFSD1 and GLMP antibodies were produced by Pineda Antikörper Service (Berlin, Germany).

### Additional antibodies used in the study

CD31 (clone MEC 13.3, BD Biosciences), vWF (A0082, DAKO), LAMP2 (clone H4B4, Developmental Studies Hybridoma Bank), Na+/K+-ATPase (clone 464.6, Millipore), VDAC (Sigma Aldrich), HA (clone 3F10; Roche), GFP (clones 7.1 and 13.1, Sigma Aldrich) Gapdh (Santa Cruz), LAMP1 (clone 1D4B, Developmental Studies Hybridoma Bank), KDEL (clone 10C3; Enzo), GM130 (Clone 35/GM130, BD Bioscience), Integrin α-IIb (clone MW Reg30, Biolegend), Limp2 (polyclonal), Cathepsin D (*Claussen et al., 1997*). All standard reagents and chemicals, if not stated otherwise, were purchased from Sigma-Aldrich.

### Generation of Mfsd1 knockout mice and genotyping

Targeted ES cells (JM8A3.N1 cell line) were obtained from the KOMP consortium (Project CSD79205, Clone EPD0728_5_G03) and provided by the UC Davis KOMP Repository. Generation, breeding and analysis of mice was in line with local and national guidelines and has been approved by the local authorities (Amt für Verbraucherschutz, Lebensmittelsicherheit und Veterinärwesen, Hamburg, No. 75/13; Ministerium für Energiewende, Landwirtschaft, Umwelt und ländliche Räume, Kiel, V 312–72241.121-3 and V 242-13648/2018). ES cells were injected into C57/BL6N cell embryos using the PiezoExpert (Eppendorf). The obtained chimeric mice were bred with C57/BL6N wild type mice to obtain germline transmission. In the resulting mouse line (tm1a, Knockout-first, *Mfsd1$^{tm1a}$ $^{(KOMP)Wtsi}$*, expression of *Mfsd1* is disrupted by a splice acceptor (SA) site in the intronic sequence between exons 2 and 3. A β-galactosidase reporter (lacZ) is expressed under the control of the endogenous *Mfsd1* promotor. To obtain the floxed allele, mice carrying the tm1a allele were bred with Flp-deleter mice. To generate conditional mice (*Mfsd1$^{tm1c(KOMP)Wtsi}$*) and cre-deleted mice in the germline (*Mfsd1$^{tm1d(KOMP)Wtsi}$*), mice were crossed with the corresponding cre-deleter mice (cre expression under the control of the Tie2- and the cytomegalovirus-promotor, respectively). Successful generation of the different intermediate alleles was confirmed by specific polymerase chain reaction (PCR). Finally, the Flp and ubiquitously expressed Cre transgenes were removed from founder mice by breeding.

### Genotyping

The genotype of mice and derived mouse embryonic fibroblasts was determined by PCR using the cycling parameters suggested by UC Davis KOMP Repository. The presence of *Mfsd1* WT, tm1a, tm1c, tm1d, flp and cre alleles in the mice genome was analyzed by PCR. The following primers were used: CSD-MFSD1-F 5'-TATGGACTCTGCCCACAGTGTTACG-3', CSD-MFSD1-ttR 5'AC TCAGCCCTTTTCTGTCTCCTACG, CSD-MFSD1-R 5'-AATGGCCAAAGACAGGCAGAAATGG-3', CSD-neoF 5'-GGGATCTCATGCTGGAGTTCTTCG-3', Cre Fw 5-ATGCGCTGGGCTCTATGGCTTC TG-3', Cre Rv 5'-TGCACACCTCCCTCTGCATGCACG-3', Flp Fw 5'-GTCACTGCAGTTTAAATACAA-GACG-3', Flp Rv 5'-GTTGCGCTAAAGAAGTATATGTGCC-3'.

## Plasmids

Constructs for mMFSD1 were cloned using mMFSD1-pcDNA6.2 cEmGFP TOPO as a template (kind gift from Bruno Gasnier, Université de Paris). mGLMP construct was cloned using Ncu-G1-His6-pcDNA3.1 (kind gift from Torben Lübke, Bielefeld University) as a template. mMFSD8 construct was cloned using mMFSD8-pEGFP-C1 (kind gift of Stephan Storch, University of Hamburg). Human hLAMP1 and mouse mLAMP1 constructs were cloned using cDNA from HeLa cells or mouse liver as a template, respectively. The mMFSD1, mGLMP and hLAMP1 cDNAs were fused to a human hemagglutinin (HA) coding sequence (protein sequence YPYDVPDYA) either at the C-terminus (MFSD1-HA, GLMP-HA and LAMP1-HA) or at the N-terminus (HA-MFSD1) by introducing the HA coding sequence in the cloning primers. For the MFSD1-HA$_{int}$ construct with an internal HA-tag between transmembrane domains 1 and 2, the HA tag was inserted between amino acids 76 and 77 with the following primers: MFSD1 (TD1-HA-TD2) Rv agcgtagtctgggacgtcgtatgggtagtcccgtttcacctgagtct and MFSD1 (TD1-HA-TD2) Fw tacccatacgacgtcccagactacgctatgcaagtgaacaccacgaa. HA coding sequence was introduced in MFSD1-HA$_{int}$ by overlap extension-PCR. The mutations MFSD1 11,12LL_AA and 195-198Y_A were introduced in the cloning primer and by site-directed mutagenesis, respectively. Primers containing restriction sites *HindIII/XbaI* (mMFSD1 and hLAMP1), *HindIII/BamHI* (mGLMP and mMFSD1), *EcoRI/BamHI* (EGFP, mGLMP and mMFSD1) and *BglII/HindIII* were purchased from Sigma-Aldrich. mMFSD1 and mGLMP were cloned into the pcDNA 3.1/Hygro (+) vector (Invitrogen), the pEGFP-N1 vector (Clontech) and the pmKATE2 vector (kind gift from Sean Froese, University of Zurich), hLAMP1 and mLAMP1 were cloned into the pcDNA 3.1/Hygro (+) vector (Invitrogen), EGFP was cloned into pmKATE2, mMFSD1 was cloned into the pEGFP-N1 vector and the constructs were sequenced by GATC Biotech (Cologne, Germany). Site directed mutagenesis was performed based on the QuickChange site-directed mutagenesis protocol with minor modifications (*Zheng et al., 2004*). LYAAT1-pcDNA6.2 was a kind gift of Stephan Storch, University of Hamburg. mLAMP1-pmKATE2 was a kind gift from Sean Froese, University of Zurich.

## Cell culture and transfection

HeLa cells were obtained from ATCC and used at early passages without further authentication. Mouse embryonic fibroblasts (MEF) were generated in the lab. No mycoplasma contamination was detected in the used cell lines. HeLa cells and mouse embryonic fibroblasts (MEFs) were cultivated in DMEM containing 4.5 g/L of D-glucose, L-glutamine (Thermo Fisher Scientific) supplemented with 10% (v/v) FBS (fetal bovine serum, Biochrom) 100 units/ml penicillin (Life-technologies) and 100 µg/ml streptomycin (Life-technologies). Wild type and *Mfsd1$^{tm1a/tm1a}$* embryos from female mice sacrificed 13.5 days post coitum were used to isolate mouse embryonic fibroblasts. The embryos were separated from the placenta and the embryonic sac. The head of the embryos was removed and used for genotyping. All red organs were dissected and the remaining embryo tissue was washed with PBS and placed in a Petri dish with 2 ml of trypsin/EDTA (0.5 mg/ml / 0.22 mg/ml in PBS). The tissue was minced with a sterile razor blade and incubated for 15 min at 37°C. The cells were collected with culture medium and centrifuged at 300 x g for 5 min at room temperature. The pellet was resuspended in 10 ml of culture medium and plated in a 10 cm culture dish. MEFs were treated with the inhibitors of lysosomal proteolysis Bafilomycin A1 (100 nM, Calbiochem), E64D (50 µM, Enzo life Biosciences) or leupeptin (50 µM, Enzo life Biosciences) for 24 hr.

For transfection of cells, 1–5 µg of DNA were incubated with polyethylenimine (PEI) in DMEM (without antibiotics nor FBS) for 15 min at room temperature. The mix was applied to the culture of cells, and after ~6 hr the media was exchanged. The transfected cells were analysed 24–48 hr post-transfection.

## Radioactive amino acid uptake assay

HEK cells were transfected with plasmid DNA. Seven hours post transfection, cells were seeded on p12 culture dishes previously coated overnight with poly-L-lysine. 24 hr post-transfection the cells were washed with Krebs-Ringer (KR) buffer (146 mM NaCl, 3 mM KCl, 1 mM CaCl$_2$, 10 mM KH$_2$PO$_4$/K$_2$HPO$_4$) at pH 7.4 and incubated for 15 min with KR buffer (pH 5.5) containing 100 µM of non-radiolabeled amino acids and 0.5 µCi of $^3$H labeled amino acids. The cells were washed with cold KR buffer (pH 7.4) on ice, lysed with 0.1 M NaOH and collected. 0.2 M KR buffer (pH 6.2) was added to the mix containing the lysed cells in order to neutralize the pH. The protein content of the cell

lysates was analyzed and the remaining cell lysates were used for radioactive measurements. Briefly, the samples were mixed with Ultima GoldTM liquid scintillation cocktail (PerkinElmer) and the disintegrations per minute (DPM) were measured in a TriCarb 2910 TR liquid scintillation counter (PerkinElmer). The radioactivity measurements were corrected by the protein concentrations.

## Subcellular fractionation

Lysosomes were isolated from the mouse liver as described previously (*Markmann et al., 2017*). Mice were injected with 4 µl/g bodyweight of a 17% (w/v in 0.9% NaCl) Triton WR1339 solution (Sigma Aldrich) three days prior to sacrifice. The liver was homogenized in five volumes of ice-cold 0.25 M sucrose with three strokes with a Potter-Elvejhem homogenizer. Homogenates were centrifuged at 1000 × g for 10 min at 4°C. The supernatant was removed and the pellet resuspended, followed by another centrifugation step at 1000 × g for 10 min at 4°C. Both supernatants were pooled (post nuclear supernatant, PNS) and centrifuged in an ultracentrifuge at 56,000 × g (70.1 Ti rotor, Beckmann, Coulter, Indianapolis, IN) for 7 min at 4°C. After re-homogenization of the pellet with 0.25 M sucrose followed by another ultracentrifugation step, the supernatant was discarded and the pellet was resuspended in sucrose solution with a density of 1.21 (resulting in the ML-fraction, representing the mitochondrial-lysosomal fraction). The ML fraction (~3.5 ml) was overlaid with 2.5 ml of sucrose solutions of 1.15, 1.14, and finally 1.06 density yielding a discontinuous sucrose gradient. The discontinuous gradient was centrifuged at 110,000 × g for 150 min in a swinging bucket rotor at 4°C (SW41 Ti rotor, Beckmann Coulter). Lysosomes/tritosomes were collected at the interphase between ρ 1.14 and 1.06 sucrose (F2-fraction).

## Magnetide beads isolation of lysosomes

Lysosomes were isolated from MEFs as described previously (*Thelen et al., 2017*). Briefly, cells were incubated with superparamagnetic nanoparticles and after a chase period, lysosomes were isolated using Miltenyi LS Separation columns in combination with a MidiMACS Magnet and successful isolation of lysosomes confirmed with an enzymatic assay for ß-Hexosaminidase activity. Isolated lysosomes were lysed using Triton X-100 and the protein concentration determined using the DC protein assay (BioRad).

## Proteomic analysis of total liver and lysosomal fractions

*Sample preparation and data acquisition:* Sample preparation and tandem mass tag (TMT) labeling was performed as described (*Koch and Dahl, 2018*). In brief, 100 µg of tritosome fractions isolated from three wildtype and *Mfsd1* KO livers were digested by in-solution digest onto centrifugal filter units. Proteins were reduced with 20 mM dithiothreitol and thiol groups were alkylated with 40 mM acrylamide before 1 µg trypsin was added for overnight digestion at 37°C. Peptides were collected in the filtrate and sodium deoxycholate was precipitated with trifluoroacetic acid. Peptides were vacuum concentrated, re-dissolved in 20 mM triethylammonium bicarbonate, and labeled with isobaric TMTsixplex reagents (ThermoFisher Darmstadt, Germany). Peptides were pooled and desalted on Oasis HLB cartridges (Waters GmbH, Eschborn, Germany). Eluates containing 70% acetonitrile, 0.1% formic acid were dried and fractionated to 12 fractions by isoelectric point with an Offgel fractionator (Agilent Technologies, Waldbronn, Germany). Peptide pool fractions were dried and stored at −20°C. Redissolved fractions were injected onto a C18 trap column (20 mm length, 100 µm inner diameter, ReproSil-Pur 120 C18-AQ, 5 µm, Dr. Maisch GmbH, Ammerbuch-Entringen, Germany) made in-house. Bound peptides were eluted onto a C18 analytical column and separated during a linear gradient from 4% to 35% solvent B (80% acetonitrile, 0.1% formic acid) within 120 min at 220 nl/min. The nanoHPLC was coupled online to an LTQ Orbitrap Velos mass spectrometer (Thermo Fisher Scientific, Bremen, Germany). Peptide ions between 330 and 1600 m/z were scanned in the Orbitrap detector with a resolution of 30,000 followed by HCD fragmentation of the 22 most intense precursor ions and detection in the Orbitrap (R = 7500). The mass spectrometry proteomics data have been deposited to the ProteomeXchange Consortium via the PRIDE partner repository with the dataset identifier PXD014241.

## Data processing

Raw data processing and analysis of database searches were performed with Proteome Discoverer software 2.1.1.21 (Thermo Fisher Scientific). Peptide identification was done with an in house Mascot server version 2.5.1 (Matrix Science Ltd, London, UK). MS2 data were searched against Swissprot database (2015_11, taxonomy: *Mus musculus)*. Precursor ion m/z tolerance was eight ppm, fragment ion tolerance 0.02 Da. Tryptic peptides with up to two missed cleavages were searched. Propiona-mide on cysteines and TMT on N-terminus and lysine were set as static modifications. Oxidation was allowed as dynamic modification of methionine, acetylation as modification of the N-terminus. Mascot results were assigned corrected p-values (q-values) by the percolator algorithm version 2.05 as implemented in Proteome Discoverer. Spectra with identifications above 0.01 q-value were sent to a second round of database search with semitryptic enzyme specificity (one missed cleavage allowed) and dynamic propionamide modification. Other parameters were the same as the first round search. Proteins were included if at least two peptides were identified with q-value <0.01. Only PSMs with co-isolation <10% and unique peptides were used for quantification.

## Quantitative realtime polymerase chain reaction (qPCR)

Total RNA was isolated using Nucleospin RNA plus kit (Macherey-Nagel) following the manufacturer´s instructions. Cells were scrapped, centrifuged at 300 x g for 5 min at 4°C and the pellet was lysed. For homogenisation of liver samples, small pieces of the organ and ceramic beads (Precellys) were added to 350 µl of lysis buffer provided by the kit and lysed using a Precellys homogeniser (Bertin Instruments) with two cycles at 6,000 rpm for 30 s. The quality of the RNA was tested by agarose gel electrophoresis. The synthesis of complementary DNA (cDNA) was performed with the RevertAid First Strand Synthesis kit according to the manufacturer´s instructions. 2 µg of total RNA and random hexamer primers (Thermo Fisher Scientific) were used for the reverse transcription.

The transcription levels of different genes were analysed by qRT-PCR using the Universal Probe Library (UPL) System Technology (Roche). The oligonucleotides and UPL probes used for every gene were designed using the Assay Design Center (Roche). Samples containing 0.5 µl of cDNA, 4.5 µl LightCycler 480 Probes Master (Roche), 0.1 µM hydrolysis probe and 0.3 µM of each oligonucleotide in a final volume of 10 µl were analysed in a Light Cycler 480 Instrument II (Roche). All the assays were performed in technical duplicates for each sample. Serial cDNA dilutions of a mixture of samples cDNA was used to calculate the primer efficiency (E=⁽10−1/slope)). ΔCp (crossing point) for each sample and gene was calculated normalizing the Cp of each assay to the Cp of the reference gene Gapdh of the same sample and assay. The results are presented relative to the the values of WT samples, considering the respective primer efficiency (RQ = E(-ddCp)).

## Deglycosylation by PNGase F/Endo H

The cell lysates were denatured in 0.5% (w/v) SDS and 250 mM ß-mercaptoethanol for 10 min at 55°C. According to the deglycosylation reaction, the samples were adjusted to a final concentration of 12 mM EDTA, 120 mM Tris HCl pH 8.0, 1.2% (w/v) CHAPS and 2 units of PNGase F or 60 mM NaAc, 0.12% (w/v) CHAPS and 5 milliunits of EndoH, incubated O/N at 37°C and analysed by Western blot.

## Protein extraction and western blotting

Cells grown on culture dishes were washed twice with PBS and 600 µl of PBS supplemented with 1x complete Protease inhibitor cocktail (Roche) were added to the plate for harvesting the cells using a cell scraper. The cell suspension was centrifuged for 8 min at 1000 x g at 4°C. The pellet was resuspended in lysis buffer (PBS, 1x complete and 1% (w/v) Triton X-100), sonicated twice for 20 s at 4°C using a Branson Sonifier 450 (level seven in a cup horn, Emerson Industrial Automation) and lysed on ice for approximately 60 min (the samples were homogenised using a vortex every 15 min). The cell suspension was centrifuged at 16,000 x g for 15 min at 4°C and the protein concentration of the supernatant was determined using the Pierce BCA (bicinchoninic Acid) Assay kit (Thermo Fisher Scientific) according to the manufacturer's instructions.

For mouse tissue homogenisation, the tissues were prepared in 20 volumes of lysis buffer and homogenised with 10–20 strokes at 1000 rounds per minute using a Glass homogenizer (B.Braun type 853202). The following steps of sonication, lysis and determination of the protein concentration of the tissue lysates was performed using the same procedure described previously for cell lysates.

Proteins were prepared in sample buffer (125 mM Tris/HCl pH6.8, 10% (v/v) glycerol, 1% (w/v) SDS, 1% (v/v) ß-mercaptoethanol and traces of bromophenol blue) and were denatured for 10 min at 55°C or 95°C depending on the hydrophobicity of the sample and its running behavior. Western blot was carried out according to standard procedures. After washing, the membranes in TBS-T buffer, horseradish peroxidase activity was detected by using an ImageQuant LAS 4000 (GE Healthcare). The intensity of the signal was quantified using Image J software. Before incubation with different antibodies, the membranes were stripped using 0.2 M NaOH. Incubations of 5 min at room temperature and gentle shaking were performed in distilled water, followed by 0.2 M NaOH, rinsing with distilled water, 0.2 M NaOH followed by distilled water, and finally TBS-T. Next, the membranes were incubated in 5% (w/v) milk powder in 1x TBS-T buffer for 1 hr at room temperature followed by incubation with first antibody.

## Crude-membrane preparation of mouse tissues/cells

Mouse tissues were homogenized in 10 volumes of homogenisation buffer (250 mM sucrose, 10 mM Tris in PBS pH 7,4 and 1x complete inhibitor) and homogenised with 10–20 strokes at 1000 rotation per minute using a Glass homogeniser (B.Braun type 853202). The lysates were centrifuged for 10 min at 1000 x g at 4°C, and the post nuclear supernatant was collected (PNS). Cells were grown in culture dishes, washed twice with PBS and collected in 600 µl of PBS supplemented with 1x complete Protease inhibitor cocktail (Roche) using a cell scraper. The cell suspension was centrifuged for 10 min at 1000 x g at 4°C and the pellet was re-suspended in homogenisation buffer. The PNS and cell suspensions were sonicated twice for 20 s at 4°C using a Branson Sonifier 450 (level seven in a cup horn, Emerson Industrial Automation) and three cycles of freeze/thaw were applied. The samples were placed into a polypropylene tube with snap-on cap (Beckmann Coulter) and ultracentrifuged at 186,000 x g in an Optima TLX Ultracentrifuge (Beckmann Coulter) using a TLA-55 rotor (Beckmann Coulter) for one hour at 4°C. The pellet was re-suspended in 2% SDS in PBS and the protein concentration was determined.

## Co-Immunoprecipitation

HeLa cells were lysed 48 hr post-transfection in immunoprecipitation buffer (1% CHAPS [3-[(3- cholamidopropyl)dimethylammonio]−1-propanesulfonate] in PBS, 120 mM NaCl, 50 mM Tris HCl, 2,5 mM $CaCl_2$, 2,5 mM $MgCl_2$ and 1x complete Protease inhibitor cocktail (Roche)). Proteins were extracted and quantified as described above and 300–1,000 µg of total protein were incubated with 1.5 µl of antibody overnight at 4°C in a rotor. At the same time, 50 µl/sample of Dynabeads protein G for Immunoprecipitation (Thermo Fisher Scientific) were blocked with 3% BSA. Next day, the beads were incubated with the lysate/antibody mixture for 2 hr at room temperature in a rotor. The samples were washed with 1 ml immunoprecipitation buffer using a magnetic separator and incubated 15 min at room temperature for three times. In a final step, the beads were incubated with 40 µl of 1x Lämmli buffer, and incubated for 10 min at 55°C or 95°C. Finally, the Co-IP samples were analyzed by SDS-PAGE and immunoblotting. HA-tagged constructs were detected with a peroxidase conjugated monoclonal antibody against HA (clone 3F10, Sigma-Aldrich). For immunoprecipitation of GFP-tagged proteins, a monoclonal mouse anti-GFP antibody (clones 7.1 and 13.1; Sigma-Aldrich) was used. For immunoblot detection of co-immunoprecipitation experiments, clean blot detection reagent was used (Thermo Fisher).

## Flow cytometry

Surface protein expression of transfected HeLa cells was analysed by flow cytometry 48 hr after transfection. Approximately $10^6$ cells were pipetted in 96-well round bottom tissue cultures plates and centrifuged at 210 x g for 5 min at 4°C. The samples were incubated with specific antibodies coupled to phycoerythrin (PE) diluted in MACS buffer (2 mM EDTA and 0.5% (w/v) BSA in PBS) for 45 min on ice in dark conditions. Next the samples were centrifuged at 210 x g for 5 min at 4°C and washed with MACS buffer. The process was repeated and the cells were resuspended in 200 µl of MACS buffer for further analysis on a FACSCanto II (BD Biosciences) device using the FACSDiva software. Positive and negative controls were used for every dye. The data was analyzed with the FlowJo 10 software.

## Flow cytometry/Fluorescence resonance energy transfer (FRET)

Flow cytometry measurements were performed using a FACSCanto II (BD Bioscience). HeLa cells were transfected with indicated plasmids containing the cDNA fused to EGFP (pEGFP-N1) or mKATE2 (pmKATE2-N) in the C-terminus, respectively. Transfected HeLa cells were excited at 488 nm 48 hr post-transfection. Voltages were adjusted with the BD FACSDiva software. GFP was excited with the 488 nm laser and measured with a 530/30 filter; any occurring FRET signal was measured with a 670 LP filter. For each sample 30.000 events were collected. The data was analyzed with FlowJo10 software.

## Indirect immunofluorescence

Semi-confluent cell cultures grown on glass coverslips were fixed for 20 min with 4% (w/v) paraformaldehyde at room temperature. Cells were permeabilized, quenched and blocked before incubation with primary antibodies overnight at 4°C. The coverslips were washed and incubated for 90 min with AlexaFluor dye-conjugated secondary antibodies (Thermo Fisher Scientific). Afterwards, the coverslips were washed four times and mounted on microscope slides with mounting medium including DAPI (4-,6-diamidino-2-phenylindole) to visualize the cell nuclei. For visualization of the samples, a FV1000 confocal laser scanning microscope (Olympus Life Science Solutions) with a U Plan S-Apo 100x oil immersion objective (NA = 1.40) was used. The images were acquired and processed with the Olympus FluoView Software.

## Immunofluorescence staining of liver sections

After mouse perfusion and post-fixation in 4% paraformaldehyde, the tissues were washed and incubated in 30% (w/v) sucrose in 0.1 M phosphate buffer for 24 hr at 4°C. Indirect immunofluorescence staining was performed on semi-thin free-floating cryosections (35 μm thick) prepared using a Leica SM 2000R sliding microtome (Leica Microsystems) and dry-ice cooling. The sections were blocked, washed and incubated with primary antibodies overnight at 4°C. After washing steps, the samples were incubated with AlexaFluor dye-conjugated secondary antibodies (Thermo Fisher Scientific) for 2 hr at room temperature under constant shaking. Finally, the sections were washed and embedded using mounting medium including DAPI. For visualization of the samples, a FV1000 confocal laser scanning microscope (Olympus Life Science Solutions) with a U Plan S-Apo 60x or 100x oil immersion objective (NA = 1.40) was used. The images were acquired and processed with the Olympus FluoView Software.

## LysoTracker Red staining

Lysotracker (LysoTracker Red DND-99, Invitrogen) staining was performed as described previously.

## X-gal staining

Semi-thin tissue cryosections were washed in PBS for 10 min shaking and incubated for 10 min in permeabilizing solution (0.01% (w/v) Na-deoxycholat, 0.02% (w/v) NP-40 in PBS). After a washing step in PBS, the samples were incubated for 3 hr in 5-bromo-4-chloro-3-indolyl-β-D-galactopyranoside (X-Gal)- solution (5 mM $K_3Fe(CN)_6$, 5 mM $K_3Fe(CN)_6$, 2 mM $MgCl_2$ and 1 mg/ml X-gal (Sigma-Aldrich)) at 37°C with gentle shaking. The sections were washed twice in PBS and embedded in mounting medium.

## Electron microscopy

Mice were perfused transcardially with 0.1 M phosphate buffer (PB) followed by 6% (v/v) glutaraldehyde (Polysciences, 00216A) in 0.1 M PB. Liver samples were post-fixed in 2% (w/v) $OsO_4$, dehydrated and embedded in Araldite M/dodecenylsuccinic anhydride. Ultrathin sections were cut on an EM UC6 ultramicrotome (Leica), collected on Formvar-coated copper grids (Agar Scientific, G2020C) and stained according to standard protocols. Images were acquired with an EM900 transmission electron microscope (Zeiss).

## Statistical analysis

Statistical analysis was performed using GraphPad Prism (GraphPad Software, San Diego, CA). For the experimental cohorts, we used 3 to 16 mice per genotype per experiment (=biological

replicates) (as indicated in the figures/figure legends for each experiment). For mouse experiments, age-matched wildtype mice from the same colony were used as controls. In cell-culture-experiments, independent dishes with independent transfections were defined as replicates. All experiments (except the proteomics experiments and determination of amino acids) were performed at least twice with the indicated number of replicates. No outliers were excluded from the data analysis. Sample size was based on our expectations on similar experiments previously performed in our laboratory, as well as in the literature. The number of animals used for each experiment is specified in the figure legends. Tests between two groups were carried out using unpaired, two-tailed student's t-test. Data are presented as mean ± standard error of the mean (SEM). Significance was labeled with one asterisk = $p < 0.05$, two asterisks = $p < 0.01$, three asterisks = $p < 0.001$, and four asterisks = $p < 0.0001$.

## Acknowledgements

We thank Maike Langer, Daniela Wiegmann, Dagmar Niemeier and Marlies Rusch for excellent technical assistance. Dirk Schmidt-Arras is acknowledged for the gift of the Tie2-cre mice. Christoph Gelhaus is acknowledged for the help with radioactivity measurements. Bruno Gasnier, Torben Lübke, Stephan Storch and Sean Froese are acknowledged for the gift of plasmids. The EUCOMM consortium is acknowledged for providing targeted ES-cells. This work was in part supported by the Deutsche Forschungsgemeinschaft (DFG) to MD (DA 1785–1) and work was funded by the Marie Curie Initial Training Network (FP7-People-2013-ITN, Grant 607446, Euroclast) to PS. The authors declare no financial and non-financial competing interests.

## Additional information

### Funding

| Funder | Grant reference number | Author |
|---|---|---|
| Deutsche Forschungsgemeinschaft | DA 1785-1 | Markus Damme |
| European Commission | 607446 Euroclast | Paul Saftig |

The funders had no role in study design, data collection and interpretation, or the decision to submit the work for publication.

### Author contributions

David Massa López, Conceptualization, Data curation, Formal analysis, Investigation, Visualization, Methodology, Writing—original draft, Writing—review and editing; Melanie Thelen, Christian Thiel, Marc Sylvester, Data curation, Formal analysis, Investigation; Felix Stahl, Arne Linhorst, Irm Hermanns-Borgmeyer, Formal analysis, Investigation; Renate Lüllmann-Rauch, Formal analysis, Supervision, Investigation; Winnie Eskild, Investigation; Paul Saftig, Funding acquisition, Investigation, Writing—review and editing; Markus Damme, Conceptualization, Formal analysis, Supervision, Funding acquisition, Validation, Investigation, Visualization, Writing—original draft, Project administration, Writing—review and editing

### Author ORCIDs

Felix Stahl http://orcid.org/0000-0002-7443-8967
Markus Damme https://orcid.org/0000-0002-9699-9351

### Ethics

Animal experimentation: Generation, breeding and analysis of mice was in line with local and national guidelines and has been approved by the local authorities (Amt für Verbraucherschutz, Lebensmittelsicherheit und Veterinärwesen, Hamburg, No. 75/13; Ministerium für Energiewende, Landwirtschaft, Umwelt und ländliche Räume, Kiel, V 312-72241.121-3 and V 242-13648/2018).

Decision letter and Author response
Decision letter https://doi.org/10.7554/eLife.50025.024
Author response https://doi.org/10.7554/eLife.50025.025

## Additional files

### Supplementary files

• Supplementary file 1. TMT-based proteomic analysis of total liver homogenates. Full list of proteins identified and quantified in three replicates of wildtype and *Mfsd1* KO mouse liver samples. Accession numbers, GenId's and normalized abundance for each protein in each replicate are given.
DOI: https://doi.org/10.7554/eLife.50025.017

• Supplementary file 2. TMT-based proteomic analysis of isolated lysosomes from liver. Full list of proteins identified and quantified in three replicates of wildtype and *Mfsd1* KO mouse liver lysosome samples. Accession numbers, GenId's and normalized abundance for each protein in each replicate are given.
DOI: https://doi.org/10.7554/eLife.50025.018

• Supplementary file 3. Key Resources Table.
DOI: https://doi.org/10.7554/eLife.50025.019

• Transparent reporting form DOI: https://doi.org/10.7554/eLife.50025.020

### Data availability

Proteomics raw-data were deposited to ProteomeXchange via the PRIDE database. Project name: MFSD1 KO Liver; project accession: PXD014241.

The following dataset was generated:

| Author(s) | Year | Dataset title | Dataset URL | Database and Identifier |
|---|---|---|---|---|
| Massa López D, Thelen M, Stahl F, Thiel C, Linhorst A, Sylvester M, Hermanns-Borgmeyer I, Luellmann-Rauch R, Eskild W, Saftig P, Damme M | 2019 | Proteomic analysis of total liver and isolated lysosomes from wildtype and MFSD1 knockout mice | https://www.ebi.ac.uk/pride/archive/projects/PXD014241 | EBI PRIDE, PXD014241 |

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
