## [Decision Letter]

Thank you for submitting your article "The lysosomal transporter MFSD1 is essential for liver homeostasis and critically depends on its accessory subunit GLMP" for consideration by *eLife*. Your article has been reviewed by three peer reviewers, one of whom served as a guest Reviewing Editor, and the evaluation has been overseen by Vivek Malhotra as the Senior Editor. The following individual involved in review of your submission has agreed to reveal their identity: Peter Lobel (Reviewer #2).

The reviewers have discussed the reviews with one another and the Reviewing Editor has drafted this decision to help you prepare a revised submission.

Summary:

This manuscript provides a thorough analysis of the MFSD1 transporter that yields new insights that represent an advance in understanding lysosome cell biology. Multiple approaches were employed to establish the lysosomal localization, lack of glycosylation and transmembrane topology of MFSD1. The authors furthermore convincingly show that MFSD1 has an important role in liver homoeostasis, phenocopies a GLMP deficiency, and that MFSD1 and GLMP are required for each other's stability. Evidence is also presented that suggests a possible interaction between MFSD1 and GLMP. These new findings will provide a valuable foundation for future studies in this field that will likely address currently unanswered questions that may include the identification of MFSD1 substrates and elucidation of the mechanisms that link MFSD1 deficiency to liver pathology. It was also noted that the lysosomal localization of the MFSD1 protein that was thoroughly established in this study contrasts with a recently published *eLife* paper that focused on functions of this transporter in the Golgi (Valoskova et al., 2019). This argues for the value of communicating the new findings presented by this new manuscript. Although all 3 reviewers expressed enthusiasm for the strengths of this manuscript, there was also consensus that some additional experiments and explanations are required to address some of the central claims of the manuscript. These are summarized below.

Essential revisions:

1) The authors claim to have found a direct physical interaction between MFSD1 and GLMP (Figure 5A). However CHAPS, the detergent used for these experiments, is known to maintain large portions of membrane attached to proteins. Thus, the apparent interaction may well be indirect. FRET is also not conclusive evidence of direct binding. Interpretation of the co-IP data should be further strengthened with the use of negative control lysosomal integral membrane proteins. Depending on the outcome of such experiments, the description of this result may need to be rephrased to indicate that the interaction isn't necessarily direct.

2) The authors should address whether or not the plasma membrane-localized dileucine mutant of MFSD1 still interacts with GLMP. Answers to this question are relevant for addressing two distinct concerns. First, this relates to the possibility that the negative results for the MFSD1 dileucine mutant transport assays could arise due to a problem with GLMP interactions at the plasma membrane. It is possible that even if MFSD1 did transport one of the substrates tested, lack of detectable transport could be due to the absence of GLMP in this system. Second, is GLMP an obligate partner for MFSD1 stabilization irrespective of the cellular compartment MFSD1 finds itself in or solely for protection from the degradative environment within lysosomes?

3) There is a lack of direct demonstration that GLMP glycosylation directly shields MFSD1 from proteases versus other alternatives such a role for GLMP in promoting a more stable conformation of MFSD1. Furthermore, GLMP stability is also highly dependent on MFSD1 which was shown not to be glycosylated. As a result, there is currently too much emphasis on the model wherein GLMP glycosylation is protective. Although the Discussion briefly acknowledges this issue, this acknowledgement is over-shadowed by emphasis on the idea that the glycosylation of GLMP is directly responsible for the protection of MFSD1. Unless more direct data can be provided in support of this concept, a more balanced presentation is required.

4) Figure 1D: along with enrichment of MFSD1 and LAMP1 in the lysosomal fractions, absence of enrichment for other organelle markers (Golgi, mitochondria, ER) should also be documented.

5) Some additional caution is required in the interpretation of the results as the authors do not identify the substrate(s) transported by MFSD1 or whether their excessive accumulation is responsible for the phenotypes observed in the MFSD1 KO animals. Although it is reasonable to predict that loss of MFSD1 transporter function is the cause of the KO phenotypes, it remains possible that loss of some other function of either MFSD1 or GLMP is also a contributing factor. These possibilities should be acknowledged and discussed.

6) Lysosome function in MFSD1 and GLMP KOs should be analyzed more carefully. Although it is implied, there is no clear demonstration of any lysosome defects. At minimum, a careful analysis of their morphology seems warranted.

---

## [Author Response]

Essential revisions:1) The authors claim to have found a direct physical interaction between MFSD1 and GLMP (Figure 5A). However CHAPS, the detergent used for these experiments, is known to maintain large portions of membrane attached to proteins. Thus, the apparent interaction may well be indirect. FRET is also not conclusive evidence of direct binding. Interpretation of the co-IP data should be further strengthened with the use of negative control lysosomal integral membrane proteins. Depending on the outcome of such experiments, the description of this result may need to be rephrased to indicate that the interaction isn't necessarily direct.

We fully agree with the reviewers that the choice of detergent for solubilization is critical for the specificity of co-immunoprecipitation experiments and that CHAPS-solubilizations retains also low-affinity interactions and might even retain unspecific interaction of proteins with unappropriated washing conditions. In order to strengthen our conclusion that the interaction between MFSD1 and GLMP is specific, we followed the suggestion of the reviewers and repeated the co-immunoprecipitation experiments with additional negative controls, both a negative control for the immunoprecipitation (LAMP1-HA) but also an additional antigen detected after immunoprecipitation (LIMP2, another lysosomal integral membrane protein).

We included LAMP1-HA already in our experiments shown in the first version of the manuscript. LAMP1-HA was expressed / precipitated with the HA antibody with lower efficiency compared to GLMP-HA and we, therefore, didn`t initially show it. However, even long exposure of the blot showed no co-immunoprecipitation of MFSD1. We now show these additional lanes in Figure 5A (left panel).

To additionally support the co-IP data also with an additional control for a possibly unspecifically bound (lysosomal integral membrane) protein (LIMP2), we repeated the co-IPs with GFP-tagged MFSD1 and HA-tagged GLMP. C-terminally GFP-tagged MFSD1 was efficiently precipitated using the GFP antibody, and HA-tagged GLMP was efficiently co-immunoprecipitated, but not HA-tagged LAMP1 included as a negative control. As an additional negative control, we detected the lysosomal membrane protein LIMP2 on the membranes after immunoprecipitation and SDS-PAGE. Endogenous LIMP2 was efficiently detected in the lysates, but no LIMP2 was detectable in the lanes after precipitation with the GFP antibody, indicating sufficient washing and providing additional evidence for specificity of the chosen conditions for MFSD1 and GLMP co-immunoprecipitation. These data are included as Figure 5—figure supplement 1B.

In summary, we think our data provide sufficient evidence to state that the conditions chosen for IP, together with FRET experiments show a specific physical interaction between MFSD1 and GLMP.

2) The authors should address whether or not the plasma membrane-localized dileucine mutant of MFSD1 still interacts with GLMP. Answers to this question are relevant for addressing two distinct concerns. First, this relates to the possibility that the negative results for the MFSD1 dileucine mutant transport assays could arise due to a problem with GLMP interactions at the plasma membrane. It is possible that even if MFSD1 did transport one of the substrates tested, lack of detectable transport could be due to the absence of GLMP in this system.

We agree with the reviewers and tested if MFSD1 and GLMP not only interact in the lysosomal membrane but also if the plasma membrane localized mutants interact by co-immunoprecipitation. For this purpose, we directed both proteins to the plasma membrane by mutating their corresponding sorting signals (dileucine-motif in MFSD1, see Figure 1, tyrosine-based sorting motif in GLMP). We could not use the MFSD1 antibody for the immunoprecipitation, since the mutation of the two critical leucines in the sorting motif of MFSD1 are located within the peptide used for immunization and the mutation renders the MFSD1^LL/AA^ protein undetectable by our antibody. We, therefore, used GFP tagged MFSD1 for this experiment. We precipitated MFSD1^LL/AA^-GFP with an antibody against GFP. HA-tagged GLMPY^400A^ was efficiently co-immunoprecipitated, though less efficient compared to the precipitation with (wildtype) MFSD1-GFP. LAMP1^Y414A^-HA, a surface localized mutant of LAMP1 used as a negative control, was not precipitated. In summary, these data show that the interaction between MFSD1^LL/AA^ and GLMP^Y400A^ not only occurs intracellularly but also at the plasma membrane. The data support the validity of these constructs for the transporter-assays. The data are included in Figure 5—figure supplement 1C.

Second, is GLMP an obligate partner for MFSD1 stabilization irrespective of the cellular compartment MFSD1 finds itself in or solely for protection from the degradative environment within lysosomes?

This is indeed an interesting and intriguing point. We tested this experimentally by using proteolytic fragmentation of MFSD1 as a readout for stabilization, which occurs in lysosomes by acid proteases (Figure 6G, Figure 6—figure supplement 1), already suggesting that GLMP confers protection specifically from lysosomal proteases. We reasoned that if GLMP would be obligate for protection or maybe even proper folding, that such fragments can be used as an indicator for stability. We overexpressed the plasma membrane localized MFSD1^LL/AA^-HA and wildtype MFSD1-HA and directly compared the levels of the C-terminal fragment. In contrast to MFSD1-HA, no C-terminal fragments were observed when MFSD1-HA is targeted to the plasma membrane, indicating that fragmentation only occurs in lysosomes and is not a result of misfolding or general destabilization. These data are now presented in Figure 6—figure supplement 1C.

3) There is a lack of direct demonstration that GLMP glycosylation directly shields MFSD1 from proteases versus other alternatives such a role for GLMP in promoting a more stable conformation of MFSD1. Furthermore, GLMP stability is also highly dependent on MFSD1 which was shown not to be glycosylated. As a result, there is currently too much emphasis on the model wherein GLMP glycosylation is protective. Although the Discussion briefly acknowledges this issue, this acknowledgement is over-shadowed by emphasis on the idea that the glycosylation of GLMP is directly responsible for the protection of MFSD1. Unless more direct data can be provided in support of this concept, a more balanced presentation is required.

Our efforts to experimentally strengthen the relevance of glycosylation of GLMP for the stability of the complex between MFSD1 and GLMP were unsuccessful. We created a GLMP cDNA in which the critical asparagine residue of all putative N-glycosylation sites (9 in total) were changed to alanine. We thought to use this construct for rescue experiments, in order to figure out if unglycosylated GLMP can still rescue MFSD1 in GLMP KO MEFs. However, this mutant was fully retained in the ER, likely due to misfolding, and could not be used for such experiments.

We therefore tried to present the data more balanced and not focusing too much on the glycosylation aspect. However, we think there is good evidence given also the cited literature in the Discussion on other pairs of interaction partners and we think that this is an important aspect that should still be covered at least in the Discussion.

Specifically, we removed the concluding sentence at the very end of the Discussion, removed the emphasis on N-glycosylation of GLMP and non-glycosylation of MFSD1 throughout the manuscript and added a part in the Discussion that the interaction might be relevant for promoting a stable conformation. The following part of the Discussion covers these aspects:

“On the other hand, while this explanation sounds reasonable for MFSD1, the levels of GLMP in Mfsd1 KO cells are also strikingly reduced, and the stability of GLMP apparently also critically depends on the non-glycosylated MFSD1, a counter-intuitive finding that implicates that not only glycosylation matters, but also the protein-protein interaction is essential to maintain a stable protease-resistant complex. It would be of great interest to determine the interaction in more detail, and which protein domains / loops of GLMP and MFSD1 convey this interaction. Additionally GLMP might be important for promoting a more stable conformation of MFSD1, and it would be interesting to figure out if the two proteins already interact in the ER. Once a natural substrate for MFSD1 is identified, a major question is of course if the interaction with GLMP is also essential for the transport, e.g. by mediating substrate recognition or increase substrate supply. Unlike channels or enzymes that are relatively static, solute carriers (to which the MFS-family belongs to) must undergo rapid conformational changes during their functions. Therefore, the presence of a stable globular protein domain may affect the conformational dynamics of these solute carriers and directly regulate its transporter activity.”

4) Figure 1D: along with enrichment of MFSD1 and LAMP1 in the lysosomal fractions, absence of enrichment for other organelle markers (Golgi, mitochondria, ER) should also be documented.

We agree that additional markers are helpful to better judge about the enrichment of lysosomes in the magnetite-isolated lysosome-enriched fractions. We therefore included immunoblots for the Golgi (GM130), ER (KDEL) and mitochondria (VDAC). These experiments show that there is some contamination of the (lysosome-enriched-) fractions with mitochondria and Golgi vesicles, which is, however, minor compared to the striking enrichment of the lysosomal marker LAMP1 (and MFSD1). We would additionally like to refer to the paper in which the method was described in detail and additional markers were tested. (Thelen et al., 2017).

5) Some additional caution is required in the interpretation of the results as the authors do not identify the substrate(s) transported by MFSD1 or whether their excessive accumulation is responsible for the phenotypes observed in the MFSD1 KO animals. Although it is reasonable to predict that loss of MFSD1 transporter function is the cause of the KO phenotypes, it remains possible that loss of some other function of either MFSD1 or GLMP is also a contributing factor. These possibilities should be acknowledged and discussed.

We agree and added a paragraph in the Discussion:

“However, while it is reasonable to assume that the accumulation of a toxic substrate or dysfunction of lysosomes due to the loss of the transporter function of MFSD1 is the underlying cause for the observed severe phenotype in Mfsd1 and Glmp KO mouse strains, we cannot provide final certainty for this hypothesis. We cannot finally exclude that GLMP might also interact with other (lysosomal membrane) proteins or provides general stability for the lysosomal membrane like LAMP proteins do. MFSD1 might be important for indirectly transporter-related functions like nutrient sensing or even transporter-unrelated processes.”

6) Lysosome function in MFSD1 and GLMP KOs should be analyzed more carefully. Although it is implied, there is no clear demonstration of any lysosome defects. At minimum, a careful analysis of their morphology seems warranted.

We agree that this is an important point. In fact we found no evidence for lysosomal storage like enlarged lysosomes or hypertrophy of lysosomal enzymes in the liver or MEFs, as typically observed under conditions of lysosomal storage, a finding that is still puzzling for us. We now included these “negative findings” in supplementary figures:

- Electron microscopy images of hepatocytes were included (Figure 2—figure supplement 1H) to reveal similar ultrastructure of lysosomes between wildtype and Mfsd1 KO mice;

- We measured the activity of three lysosomal hydrolases to test for hyperactivity / hypertrophy of the lysosomal system (Figure 2—figure supplement 1I);

- We stained MEFs for LAMP1, cathepsin D and LysoTracker in order to qualitatively test for acidification of lysosomes, delivery of cathepsin D to lysosomes and lysosomal size in MEFs of Mfsd1 and Glmp KO mice (Figure 6—figure supplement 1A).

Additionally we, highlighted in the Discussion that we did *not* find any evidence for lysosomal storage.

Discussion:

“Our major goal of this study was to identify the substrate(s) of MFSD1 by analyzing knockout mice, assuming that a substrate accumulates in lysosomes. The phenotype of Mfsd1 KO mice is complex and the liver phenotype did not resemble that observed in mouse models of lysosomal storage diseases. We failed to detect any signs of typical lysosomal storage like aberrant storage material or enlarged lysosomes in the liver or MEFs.”